# Dirichlet-based Uncertainty Calibration for Active Domain Adaptation

**Mixue Xie, Shuang Li✉, Rui Zhang, Chi Harold Liu**
Beijing Institute of Technology, China
{mxxie,shuangli,zhangrui20}@bit.edu.cn, liuchi02@gmail.com

## Abstract

Active domain adaptation (DA) aims to maximally boost the model adaptation on a new target domain by actively selecting limited target data to annotate, whereas traditional active learning methods may be less effective since they do not consider the domain shift issue. Despite active DA methods address this by further proposing targetness to measure the representativeness of target domain characteristics, their predictive uncertainty is usually based on the prediction of deterministic models, which can easily be miscalibrated on data with distribution shift. Considering this, we propose a *Dirichlet-based Uncertainty Calibration* (DUC) approach for active DA, which simultaneously achieves the mitigation of miscalibration and the selection of informative target samples. Specifically, we place a Dirichlet prior on the prediction and interpret the prediction as a distribution on the probability simplex, rather than a point estimate like deterministic models. This manner enables us to consider all possible predictions, mitigating the miscalibration of unilateral prediction. Then a two-round selection strategy based on different uncertainty origins is designed to select target samples that are both representative of target domain and conducive to discriminability. Extensive experiments on cross-domain image classification and semantic segmentation validate the superiority of DUC.

## 1 Introduction

Despite the superb performances of deep neural networks (DNNs) on various tasks (Krizhevsky et al., 2012; Chen et al., 2015), their training typically requires massive annotations, which poses formidable cost for practical applications. Moreover, they commonly assume training and testing data follow the same distribution, making the model brittle to distribution shifts (Ben-David et al., 2010). Alternatively, unsupervised domain adaptation (UDA) has been widely studied, which assists the model learning on an unlabeled target domain by transferring the knowledge from a labeled source domain (Ganin & Lempitsky, 2015; Long et al., 2018). Despite the great advances of UDA, the unavailability of target labels greatly limits its performance, presenting a huge gap with the supervised counterpart. Actually, given an acceptable budget, a small set of target data can be annotated to significantly boost the performance of UDA. With this consideration, recent works (Fu et al., 2021; Prabhu et al., 2021) integrate the idea of active learning (AL) into DA, resulting in active DA.

The core of active DA is to annotate the most valuable target samples for maximally benefiting the adaptation. However, traditional AL methods based on either predictive uncertainty or diversity are less effective for active DA, since they do not consider the domain shift. For predictive uncertainty (e.g., margin (Joshi et al., 2009), entropy (Wang & Shang, 2014)) based methods, they cannot measure the target-representativeness of samples. As a result, the selected samples are often redundant and less informative. As for diversity based methods (Sener & Savarese, 2018; Nguyen & Smeulders, 2004), they may select samples that are already well-aligned with source domain (Prabhu et al., 2021). Aware of these, active DA methods integrate both predictive uncertainty and targetness into the selection process (Su et al., 2019; Fu et al., 2021; Prabhu et al., 2021). Yet, existing focus is on the measurement of targetness, e.g., using domain discriminator (Su et al., 2019) or clustering (Prabhu et al., 2021). The predictive uncertainty they used is still mainly based on the prediction of deterministic models, which is essentially a point estimate (Sensoy et al., 2018) and can easily be miscalibrated on data with distribution shift (Guo et al., 2017). As in Fig. 1(a), standard DNN is wrongly overconfident on most target data. Correspondingly, its predictive uncertainty is unreliable.

(a) Standard DNN v.s. Dirichlet-based Model      (b) Examples of different prediction distributions

Figure 1: (a): point-estimate entropy of DNN and expected entropy of Dirichlet-based model, where colors of points denote class identities. Both models are trained with source data. (b): examples of the prediction distribution of three "monitor" images on the simplex. The model is trained with images of "keyboard", "computer" and "monitor" from the *Clipart* domain of Office-Home dataset. For the two images from the *Real-World* domain, the entropy of expected prediction cannot distinguish them, whereas $U_{dis}$ and $U_{data}$ calculated based on the prediction distribution can reflect what contributes more to their uncertainty and be utilized to guarantee the information diversity of selected data.

To solve this, we propose a *Dirichlet-based Uncertainty Calibration* (DUC) method for active DA, which is mainly built on the Dirichlet-based evidential deep learning (EDL) (Sensoy et al., 2018). In EDL, a Dirichlet prior is placed on the class probabilities, by which the prediction is interpreted as a distribution on the probability simplex. That is, the prediction is no longer a point estimate and each prediction occurs with a certain probability. The resulting benefit is that the miscalibration of unilateral prediction can be mitigated by considering all possible predictions. For illustration, we plot the expected entropy of all possible predictions using the Dirichlet-based model in Fig. 1(a). And we see that most target data with domain shift are calibrated to have greater uncertainty, which can avoid the omission of potentially valuable target samples in deterministic model based-methods.

Besides, based on Subjective Logic (Jøsang, 2016), the Dirichlet-based evidential model intrinsically captures different origins of uncertainty: the lack of evidences and the conflict of evidences. This property further motivates us to consider different uncertainty origins during the process of sample selection, so as to comprehensively measure the value of samples from different aspects. Specifically, we introduce the distribution uncertainty to express the lack of evidences, which mainly arises from the distribution mismatch, i.e., the model is unfamiliar with the data and lacks knowledge about it. In addition, the conflict of evidences is expressed as the data uncertainty, which comes from the natural data complexity, e.g., low discriminability. And the two uncertainties are respectively captured by the spread and location of the Dirichlet distribution on the probability simplex. As in Fig. 1(b), the real-world style of the first target image obviously differs from source domain and presents a broader spread on the probability simplex, i.e., higher distribution uncertainty. This uncertainty enables us to measure the targetness without introducing the domain discriminator or clustering, greatly saving computation costs. While the second target image provides different information mainly from the aspect of discriminability, with the Dirichlet distribution concentrated around the center of the simplex. Based on the two different origins of uncertainty, we design a two-round selection strategy to select both target-representative and discriminability-conducive samples for label query.

**Contributions: 1)** We explore the uncertainty miscalibration problem that is ignored by existing active DA methods, and achieve the informative sample selection and uncertainty calibration simultaneously within a unified framework. **2)** We provide a novel perspective for active DA by introducing the Dirichlet-based evidential model, and design an uncertainty origin-aware selection strategy to comprehensively evaluate the value of samples. Notably, no domain discriminator or clustering is used, which is more elegant and saves computation costs. **3)** Extensive experiments on both cross-domain image classification and semantic segmentation validate the superiority of our method.

## 2 RELATED WORK

**Active Learning (AL)** aims to reduce the labeling cost by querying the most informative samples to annotate (Ren et al., 2022), and the core of AL is the query strategy for sample selection. Committee-based strategy selects samples with the largest prediction disagreement between multiple classifiers (Seung et al., 1992; Dagan & Engelson, 1995). Representative-based strategy chooses a set of representative samples in the latent space by clustering or core-set selection (Nguyen & Smeulders,

2004; Sener & Savarese, 2018). Uncertainty-based strategy picks samples based on the prediction confidence (Lewis & Catlett, 1994), entropy (Wang & Shang, 2014; Huang et al., 2018), etc, to annotate samples that the model is most uncertain about. Although these query strategies have shown promising performances, traditional AL usually assumes that the labeled data and unlabeled data follow the same distribution, which may not well deal with the domain shift in active DA.

**Active Learning for Domain Adaptation** intends to maximally boost the model adaption from source to target domain by selecting the most valuable target data to annotate, given a limited labeling budget. With the limitation of traditional AL, researchers incorporate AL with additional criteria of targetness (i.e., the representativeness of target domain). For instance, besides predictive uncertainty, AADA (Su et al., 2019) and TQS (Fu et al., 2021) additionally use the score of domain discriminator to represent targetness. Yet, the learning of domain discriminator is not directly linked with the classifier, which may cause selected samples not necessarily beneficial for classification. Another line models targetness based on clustering, e.g., CLUE (Prabhu et al., 2021) and DBAL (Deheeger et al., 2021). Differently, EADA (Xie et al., 2021) represents targetness as free energy bias and explicitly reduces the free energy bias across domains to mitigate the domain shift. Despite the advances, the focus of existing active DA methods is on the measurement of targetness. Their predictive uncertainty is still based on the point estimate of prediction, which can easily be miscalibrated on target data.

**Deep Learning Uncertainty** measures the trustworthiness of decisions from DNNs. One line of the research concentrates on better estimating the predictive uncertainty of deterministic models via ensemble (Lakshminarayanan et al., 2017) or calibration (Guo et al., 2017). Another line explores to combine deep learning with Bayesian probability theory (Denker & LeCun, 1990; Goan & Fookes, 2020). Despite the potential benefits, BNNs are limited by the intractable posterior inference and expensive sampling for uncertainty estimation (Amini et al., 2020). Recently, evidential deep learning (EDL) (Sensoy et al., 2018) is proposed to reason the uncertainty based on the belief or evidence theory (Dempster, 2008; Jøsang, 2016), where the categorical prediction is interpreted as a distribution by placing a Dirichlet prior on the class probabilities. Compared with BNNs which need multiple samplings to estimate the uncertainty, EDL requires only a single forward pass, greatly saving computational costs. Attracted by the benefit, TNT (Chen et al., 2022) leverages it for detecting novel classes, GKDE Zhao et al. (2020) integrates it into graph neural networks for detecting out-of-distribution nodes, and TCL (Li et al., 2022) utilizes it for trustworthy long-tailed classification. Yet, researches on how to effectively use EDL for active DA remain scarce.

## 3 DIRICHLET-BASED UNCERTAINTY CALIBRATION FOR ACTIVE DA

### 3.1 PROBLEM FORMULATION

Formally, in active DA, there are a labeled source domain $\mathcal{S} = \{\boldsymbol{x}_i^s, y_i^s\}_{i=1}^{n_s}$ and an unlabeled target domain $\mathcal{T} = \{\boldsymbol{x}_j^t\}_{j=1}^{n_t}$, where $y_i^s \in \{1, 2, \cdots, C\}$ is the label of source sample $\boldsymbol{x}_i^s$ and $C$ is the number of classes. Following the standard setting in (Fu et al., 2021), we assume that source and target domains share the same label space $\mathcal{Y} = \{1, 2, \cdots, C\}$ but follow different data distributions. Meanwhile, we denote a labeled target set as $\mathcal{T}^l$, which is an empty set $\varnothing$ initially. When training reaches the active selection step, $b$ unlabeled target samples will be selected to query their labels from the oracle and added into $\mathcal{T}^l$. Then we have $\mathcal{T} = \mathcal{T}^l \cup \mathcal{T}^u$, where $\mathcal{T}^u$ is the remaining unlabeled target set. Such active selection step repeats several times until reaching the total labeling budget $B$.

To get maximal benefit from limited labeling budget, the main challenge of active DA is how to select the most valuable target samples to annotate under the domain shift, which has been studied by several active DA methods (Su et al., 2019; Fu et al., 2021; Prabhu et al., 2021; Deheeger et al., 2021; Xie et al., 2021). Though they have specially considered targetness to represent target domain characteristics, their predictive uncertainty is still mainly based on the prediction of deterministic models, which can easily be miscalibrated under the domain shift, as found in (Lakshminarayanan et al., 2017; Guo et al., 2017). Instead, we tackle active DA via the Dirichlet-based evidential model, which treats categorical prediction as a distribution rather than a point estimate like previous methods.

### 3.2 PRELIMINARY OF DIRICHLET-BASED EVIDENTIAL MODEL

Let us start with the general $C$-class classification. $\mathcal{X}$ denotes the input space and the deep model $f$ parameterized with $\boldsymbol{\theta}$ maps the instance $\boldsymbol{x} \in \mathcal{X}$ into a $C$-dimensional vector, i.e., $f : \mathcal{X} \to \mathbb{R}^C$. For

standard DNN, the softmax operator is usually adopted on the top of $f$ to convert the logit vector into the prediction of class probability vector $\boldsymbol{\rho}^1$, while this manner essentially gives a point estimate of $\boldsymbol{\rho}$ and can easily be miscalibrated on data with distribution shift (Guo et al., 2017).

To overcome this, Dirichlet-based evidential model is proposed by Sensoy et al. (2018), which treats the prediction of class probability vector $\boldsymbol{\rho}$ as the generation of subjective opinions. And each subjective opinion appears with certain degrees of uncertainty. In other words, unlike traditional DNNs, evidential model treats $\boldsymbol{\rho}$ as a random variable. Specifically, a Dirichlet distribution, the conjugate prior distribution of the multinomial distribution, is placed over $\boldsymbol{\rho}$ to represent the probability density of each possible $\boldsymbol{\rho}$. Given sample $\boldsymbol{x}_i$, the probability density function of $\boldsymbol{\rho}$ is denoted as

$$p(\boldsymbol{\rho}|\boldsymbol{x_i},\boldsymbol{\theta}) = Dir(\boldsymbol{\rho}|\boldsymbol{\alpha}_i) = \begin{cases} \frac{\Gamma(\sum_{c=1}^{C}\alpha_{ic})}{\prod_{c=1}^{C}\Gamma(\alpha_{ic})}\prod_{c=1}^{C}\rho_c^{\alpha_{ic}-1}, & if\ \boldsymbol{\rho}\in\triangle^C \\ 0 & ,\ otherwise \end{cases},\quad \alpha_{ic}>0, \quad (1)$$

where $\boldsymbol{\alpha}_i$ is the parameters of the Dirichlet distribution for sample $\boldsymbol{x}_i$, $\Gamma(\cdot)$ is the Gamma function and $\triangle^C$ is the $C$-dimensional unit simplex: $\triangle^C = \{\boldsymbol{\rho}|\sum_{c=1}^{C}\rho_c = 1 \text{ and } \forall\rho_c, 0\leq\rho_c\leq 1\}$. For $\boldsymbol{\alpha}_i$, it can be expressed as $\boldsymbol{\alpha}_i = g(f(\boldsymbol{x}_i,\boldsymbol{\theta}))$, where $g(\cdot)$ is a function (e.g., exponential function) to keep $\boldsymbol{\alpha}_i$ positive. In this way, the prediction of each sample is interpreted as a distribution over the probability simplex, rather than a point on it. And we can mitigate the uncertainty miscalibration by considering all possible predictions rather than unilateral prediction.

Further, based on the theory of Subjective Logic (Jøsang, 2016) and DST (Dempster, 2008), the parameters $\boldsymbol{\alpha}_i$ of Dirichlet distribution is closely linked with the evidences collected to support the subjective opinion for sample $\boldsymbol{x}_i$, via the equation $\boldsymbol{e}_i = \boldsymbol{\alpha}_i - \boldsymbol{1}$ where $\boldsymbol{e}_i$ is the evidence vector. And the uncertainty of each subjective opinion $\boldsymbol{\rho}$ also relates to the collected evidences. Both the lack of evidences and the conflict of evidences can result in uncertainty. Having the relation between $\boldsymbol{\alpha}_i$ and evidences, the two origins of uncertainty are naturally reflected by the different characteristics of Dirichlet distribution: the spread and the location over the simplex, respectively. As shown in Fig. 1(b), opinions with lower amount of evidences have broader spread on the simplex, while the opinions with conflicting evidences locate close to the center of the simplex and present low discriminability.

**Connection with softmax-based DNNs.** Considering sample $\boldsymbol{x}_i$, the predicted probability for class $c$ can be denoted as Eq. (2), by marginalizing over $\boldsymbol{\rho}$. The derivation is in Sec. E.1 of the appendix.

$$P(y=c|\boldsymbol{x}_i,\boldsymbol{\theta}) = \int p(y=c|\boldsymbol{\rho})p(\boldsymbol{\rho}|\boldsymbol{x}_i,\boldsymbol{\theta})d\boldsymbol{\rho} = \frac{\alpha_{ic}}{\sum_{k=1}^{C}\alpha_{ik}} = \frac{g(f_c(\boldsymbol{x}_i,\boldsymbol{\theta}))}{\sum_{k=1}^{C}g(f_k(\boldsymbol{x}_i,\boldsymbol{\theta}))} = \mathbb{E}[Dir(\rho_c|\boldsymbol{\alpha}_i)]. \quad (2)$$

Specially, if $g(\cdot)$ adopts the exponential function, then softmax-based DNNs can be viewed as predicting the expectation of Dirichlet distribution. However, the marginalization process will conflate uncertainties from different origins, making it hard to ensure the information diversity of selected samples, because we do not know what information the sample can bring.

### 3.3 SELECTION STRATEGY WITH AWARENESS OF UNCERTAINTY ORIGINS

In active DA, to gain the utmost benefit from limited labeling budget, the selected samples ideally should be 1) representative of target distribution and 2) conducive to discriminability. For the former, existing active DA methods either use the score of domain discriminator (Su et al., 2019; Fu et al., 2021) or the distance to cluster centers (Prabhu et al., 2021; Deheeger et al., 2021). As for the latter, predictive uncertainty (e.g., margin (Xie et al., 2021), entropy (Prabhu et al., 2021)) of standard DNNs is utilized to express the discriminability of target samples. Differently, we denote the two characteristics in a unified framework, without introducing domain discriminator or clustering.

For the evidential model supervised with source data, if target samples are obviously distinct from source domain, e.g., the realistic v.s. clipart style, the evidences collected for these target samples may be insufficient, because the model lacks the knowledge about this kind of data. Built on this, we use the uncertainty resulting from the lack of evidences, called distribution uncertainty, to measure the targetness. Specifically, the distribution uncertainty $U_{dis}$ of sample $\boldsymbol{x}_j$ is defined as

$$U_{dis}(\boldsymbol{x}_j,\boldsymbol{\theta}) \triangleq I[y,\boldsymbol{\rho}|\boldsymbol{x}_j,\boldsymbol{\theta}] = \sum_{c=1}^{C}\bar{\rho}_{jc}\left(\psi(\alpha_{jc}+1)-\psi(\sum_{k=1}^{C}\alpha_{jk}+1)\right) - \sum_{c=1}^{C}\bar{\rho}_{jc}\log\bar{\rho}_{jc}, \quad (3)$$

---

$^1\boldsymbol{\rho} = [\rho_1,\rho_2,\cdots,\rho_C]^\top = [P(y=1),P(y=2),\cdots,P(y=C)]^\top$ is a vector of class probabilities.

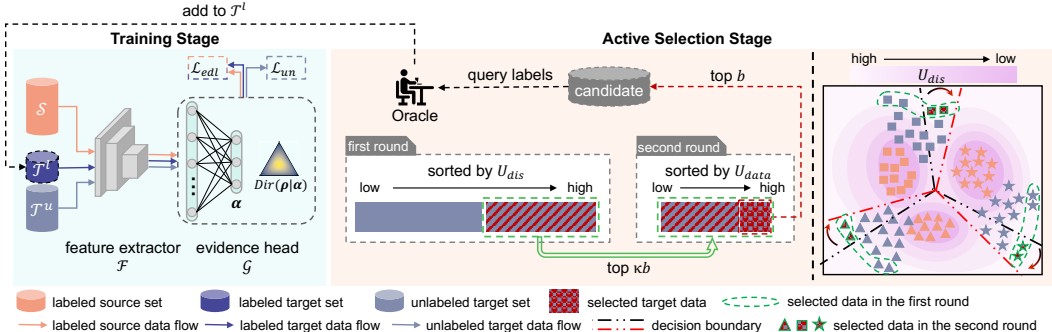

Figure 2: Illustration of DUC. When the training reaches the active selection steps, the distribution uncertainty $U_{dis}$ and data uncertainty $U_{data}$ of unlabeled target samples are calculated according to the Dirichlet distribution with parameter $\boldsymbol{\alpha}$. Then $\kappa b$ samples with the highest $U_{dis}$ are chosen in the first round. In the second round, according to $U_{data}$, we select the top $b$ samples from the instances chosen in the first round to query their labels. These labeled target samples are added into the supervised learning. When reaching the total labeling budget $B$, the active selection stops.

where $\boldsymbol{\theta}$ is the parameters of the evidential deep model, $\psi(\cdot)$ is the digamma function and $\bar{\rho}_{jc} = \mathbb{E}[Dir(\rho_c|\boldsymbol{\alpha}_j)]$. Here, we use mutual information to measure the spread of Dirichlet distribution on the simplex like Malinin & Gales (2018). The higher $U_{dis}$ indicates larger variance of opinions due to the lack of evidences, i.e., the Dirichlet distribution is broadly spread on the probability simplex.

For the discriminability, we also utilize the predictive entropy to quantify. But different from previous methods which are based on the point estimate (i.e., the expectation of Dirichlet distribution), we denote it as the expected entropy of all possible predictions. Specifically, given sample $\boldsymbol{x}_j$ and model parameters $\boldsymbol{\theta}$, the data uncertainty $U_{data}$ is expressed as

$$U_{data}(\boldsymbol{x}_j, \boldsymbol{\theta}) \triangleq \mathbb{E}_{p(\boldsymbol{\rho}|\boldsymbol{x}_j, \boldsymbol{\theta})}[H[P(y|\boldsymbol{\rho})]] = \sum_{c=1}^{C} \bar{\rho}_{jc}\left(\psi(\sum_{k=1}^{C}\alpha_{jk}+1) - \psi(\alpha_{jc}+1)\right). \quad (4)$$

Here, we do not adopt $H[\mathbb{E}[Dir(\boldsymbol{\rho}|\boldsymbol{\alpha}_j)]]$, i.e., the entropy of point estimate, to denote data uncertainty, in that the expectation operation will conflate uncertainties from different origins as shown in Eq. (2).

Having the distribution uncertainty $U_{dis}$ and data uncertainty $U_{data}$, we select target samples according to the strategy in Fig. 2. In each active selection step, we select samples in two rounds. In the first round, top $\kappa b$ target samples with highest $U_{dis}$ are selected. Then according to data uncertainty $U_{data}$, we choose the top $b$ target samples from the candidates in the first round to query labels. Experiments on the uncertainty ordering and selection ratio in the first round are provided in Sec. D.1 and Sec. 4.3.

**Relation between $U_{dis}, U_{data}$ and typical entropy.** Firstly, according to Eq. (2), the typical entropy of sample $\boldsymbol{x}_j$ can be denoted as $H[P(y|\boldsymbol{x}_j, \boldsymbol{\theta})] = H[\mathbb{E}[Dir(\boldsymbol{\rho}|\boldsymbol{\alpha}_j)]] = -\sum_{c=1}^{C}\bar{\rho}_{jc}\log\bar{\rho}_{jc}$, where $\bar{\rho}_{jc} = \mathbb{E}[Dir(\rho_c|\boldsymbol{\alpha}_j)]$. Then we have $U_{dis}(\boldsymbol{x}_j, \boldsymbol{\theta}) + U_{data}(\boldsymbol{x}_j, \boldsymbol{\theta}) = H[P(y|\boldsymbol{x}_j, \boldsymbol{\theta})]$, by adding Eq. (3) and Eq. 4 together. We can see that our method actually equals to decomposing the typical entropy into two origins of uncertainty, by which our selection criteria are both closely related to the prediction. While the targetness measured with domain discriminator or clustering centers is not directly linked with the prediction, and thus the selected samples may already be nicely classified.

**Discussion.** Although Malinin & Gales (2018) propose Dirichlet Prior Network (DPN) to distinguish between data and distribution uncertainty, their objective differs from us. Malinin & Gales (2018) mainly aims to detect out-of-distribution (OOD) data, and DPN is trained using the KL-divergence between the model and the ground-truth Dirichlet distribution. Frustratingly, the ground-truth Dirichlet distribution is unknown. Though they manually construct a Dirichlet distribution as the proxy, the parameter of Dirichlet for the ground-truth class still needs to be set by hand, rather than learned from data. In contrast, by interpreting from an evidential perspective, our method does not require the ground-truth Dirichlet distribution and automatically learns sample-wise Dirichlet distribution by maximizing the evidence of ground-truth class and minimizing the evidences of wrong classes, which is shown in Sec. 3.4. Besides, they expect to generate a flat Dirichlet distribution for OOD data, while this is not desired on our target data, since our goal is to improve their accuracy. Hence, we additionally introduce losses to reduce the distribution and data uncertainty of target data.

## 3.4 Evidential Model Learning

To get reliable and consistent opinions for labeled data, the evidential model is trained to generate sharp Dirichlet distribution located at the corner of the simlpex for these labeled data. Concretely, we train the model by minimizing the negative logarithm of the marginal likelihood ($\mathcal{L}_{nll}$) and the KL-divergence between two Dirichlet distributions ($\mathcal{L}_{KL}$). $\mathcal{L}_{nll}$ is expressed as

$$\mathcal{L}_{nll} = \frac{1}{n_s} \sum_{\boldsymbol{x}_i \in \mathcal{S}} -\log\left(\int p(y=y_i|\boldsymbol{\rho})p(\boldsymbol{\rho}|\boldsymbol{x}_i,\boldsymbol{\theta})d\boldsymbol{\rho}\right) + \frac{1}{|\mathcal{T}^l|} \sum_{\boldsymbol{x}_j \in \mathcal{T}^l} -\log\left(\int p(y=y_j|\boldsymbol{\rho})p(\boldsymbol{\rho}|\boldsymbol{x}_j,\boldsymbol{\theta})d\boldsymbol{\rho}\right)$$

$$= \frac{1}{n_s} \sum_{\boldsymbol{x}_i \in \mathcal{S}} \sum_{c=1}^{C} \Upsilon_{ic}\left(\log(\sum_{c=1}^{C}\alpha_{ic}) - \log\alpha_{ic}\right) + \frac{1}{|\mathcal{T}^l|} \sum_{\boldsymbol{x}_j \in \mathcal{T}^l} \sum_{c=1}^{C} \Upsilon_{jc}\left(\log(\sum_{c=1}^{C}\alpha_{jc}) - \log\alpha_{jc}\right), \quad (5)$$

where $\Upsilon_{ic}/\Upsilon_{jc}$ is the $c$-th element of the one-hot label vector $\boldsymbol{\Upsilon}_i/\boldsymbol{\Upsilon}_j$ of sample $\boldsymbol{x}_i/\boldsymbol{x}_j$. $\mathcal{L}_{nll}$ is minimized to ensure the correctness of prediction. As for $\mathcal{L}_{kl}$, it is denoted as

$$\mathcal{L}_{kl} = \frac{1}{C \cdot n_s} \sum_{\boldsymbol{x}_i \in \mathcal{S}} KL\left[Dir(\boldsymbol{\rho}|\tilde{\boldsymbol{\alpha}}_i)\|Dir(\boldsymbol{\rho}|\mathbf{1})\right] + \frac{1}{C \cdot |\mathcal{T}^l|} \sum_{\boldsymbol{x}_j \in \mathcal{T}^l} KL\left[Dir(\boldsymbol{\rho}|\tilde{\boldsymbol{\alpha}}_j)\|Dir(\boldsymbol{\rho}|\mathbf{1})\right], \quad (6)$$

where $\tilde{\boldsymbol{\alpha}}_{i/j} = \boldsymbol{\Upsilon}_{i/j} + (\mathbf{1} - \boldsymbol{\Upsilon}_{i/j}) \odot \boldsymbol{\alpha}_{i/j}$ and $\odot$ is the element-wise multiplication. $\tilde{\boldsymbol{\alpha}}_{i/j}$ can be seen as removing the evidence of ground-truth class. Minimizing $\mathcal{L}_{kl}$ will force the evidences of other classes to reduce, avoiding the collection of mis-leading evidences and increasing discriminability. Here, we divide the KL-divergence by the number of classes, since its scale differs largely for different $C$. Due to the space limitation, the computable expression is given in Sec. E.4 of the appendix.

In addition to the training on the labeled data, we also explicitly reduce the distribution and data uncertainties of unlabeled target data by minimizing $\mathcal{L}_{un}$, which is formulated as

$$\mathcal{L}_{un} = \beta\mathcal{L}_{U_{dis}} + \lambda\mathcal{L}_{U_{data}} = \frac{\beta}{|\mathcal{T}^u|} \sum_{\boldsymbol{x}_k \in \mathcal{T}^u} U_{dis}(\boldsymbol{x}_k,\boldsymbol{\theta}) + \frac{\lambda}{|\mathcal{T}^u|} \sum_{\boldsymbol{x}_k \in \mathcal{T}^u} U_{data}(\boldsymbol{x}_k,\boldsymbol{\theta}), \quad (7)$$

where $\beta$ and $\lambda$ are two hyper-parameters to balance the two losses. On the one hand, this regularizer term is conducive to improving the predictive confidence of some target samples. On the other hand, it contributes to selecting valuable samples, whose uncertainty can not be easily reduced by the model itself and external annotation is needed to provide more guidance. To sum up, the total training loss is

$$\mathcal{L}_{total} = \mathcal{L}_{edl} + \mathcal{L}_{un} = (\mathcal{L}_{nll} + \mathcal{L}_{kl}) + (\beta\mathcal{L}_{U_{dis}} + \lambda\mathcal{L}_{U_{data}}). \quad (8)$$

The training procedure of DUC is shown in Sec. B of the appendix. And for the inference stage, we simply use the expected opinion, i.e., the expectation of Dirichlet distribution, as the final prediction.

**Discussion.** Firstly, $\mathcal{L}_{nll}, \mathcal{L}_{kl}$ are actually widely used in EDL-inspired methods for supervision, e.g., Bao et al. (2021); Chen et al. (2022); Li et al. (2022). Secondly, our motivation and methodology differs from EDL. EDL does not consider the origin of uncertainty, since it is mainly proposed for OOD detection, which is less concerned with that. And models can reject samples as long as the total uncertainty is high. By contrast, our goal is to select the most valuable target samples for model adaption. Though target samples can be seen as OOD samples to some extent, simply sorting them by the total uncertainty is not a good strategy, since the total uncertainty can not reflect the diversity of information. A better choice is to measure the value of samples from multiple aspects. Hence, we introduce a two-round selection strategy based on different uncertainty origins. Besides, according to the results in Sec. D.5, our method can empirically mitigate the domain shift by minimizing $\mathcal{L}_{U_{dis}}$, which makes our method more suitable for active DA. Comparatively, this is not included in EDL.

**Time Complexity of Query Selection.** The consumed time in the selection process mainly comes from the sorting of samples. In the first round of each active section step, the time complexity is $\mathcal{O}(|\mathcal{T}^u|\log|\mathcal{T}^u|)$. And in the second round, the complexity is $\mathcal{O}((\kappa b)\log(\kappa b))$. Thus, the complexity of each selection step is $\mathcal{O}(|\mathcal{T}^u|\log|\mathcal{T}^u|) + \mathcal{O}((\kappa b)\log(\kappa b))$. Assuming the number of total selection steps is $r$, then the total complexity is $\sum_{m=1}^{r}(\mathcal{O}(|\mathcal{T}_m^u|\log|\mathcal{T}_m^u|) + \mathcal{O}((\kappa b)\log(\kappa b)))$, where $\mathcal{T}_m^u$ is the unlabeled target set in the $m$-th active selection step. Since $r$ is quite small (5 in our paper), and $\kappa b \leq |\mathcal{T}_m^u| \leq n_t$, the approximated time complexity is denoted as $\mathcal{O}(n_t\log n_t)$.

Table 1: Accuracy (%) on miniDomainNet with 5% target samples as the labeling budget (ResNet-50).

| Method | clp→pnt | clp→rel | clp→skt | pnt→clp | pnt→rel | pnt→skt | rel→clp | rel→pnt | rel→skt | skt→clp | skt→pnt | skt→rel | Avg |
|---|---|---|---|---|---|---|---|---|---|---|---|---|---|
| Source-only | 52.1 | 63.0 | 49.4 | 55.9 | 73.0 | 51.1 | 56.8 | 61.0 | 50.0 | 54.0 | 48.9 | 60.3 | 56.3 |
| Random | 61.6 | 78.7 | 61.6 | 64.0 | 78.7 | 63.7 | 60.5 | 64.3 | 61.1 | 64.8 | 58.7 | 75.2 | 66.1 |
| BvSB (Joshi et al., 2009) | 63.2 | 77.9 | 62.7 | 66.7 | 80.5 | 64.9 | 64.3 | 67.0 | 62.2 | 67.6 | 62.5 | 77.8 | 68.1 |
| Entropy (Wang & Shang, 2014) | 63.3 | 78.3 | 61.0 | 65.7 | 81.4 | 63.2 | 63.3 | 66.2 | 63.0 | 67.9 | 60.5 | 78.3 | 67.7 |
| CoreSet (Sener & Savarese, 2018) | 62.6 | 78.3 | 60.2 | 62.1 | 79.9 | 63.6 | 63.6 | 65.2 | 59.1 | 63.1 | 62.3 | 78.1 | 66.5 |
| WAAL (Shui et al., 2020) | 63.2 | 80.2 | 62.1 | 60.6 | 80.3 | 64.6 | 62.9 | 64.1 | 59.5 | 65.4 | 61.8 | 78.6 | 66.9 |
| BADGE (Ash et al., 2020) | 64.3 | 80.8 | 63.5 | 65.2 | 80.2 | 63.8 | 65.9 | 65.4 | 63.4 | 66.7 | 63.3 | 79.2 | 68.5 |
| AADA (Su et al., 2019) | 62.4 | 77.5 | 61.7 | 61.9 | 79.7 | 61.1 | 65.6 | 66.0 | 60.8 | 65.1 | 62.1 | 80.0 | 67.0 |
| DBAL (Deheeger et al., 2021) | 62.9 | 79.2 | 60.8 | 64.6 | 78.1 | 62.5 | 65.6 | 65.2 | 59.2 | 66.3 | 61.3 | 80.3 | 67.2 |
| TQS (Fu et al., 2021) | **67.8** | **82.0** | 65.4 | 67.5 | **84.8** | 66.1 | 63.8 | 67.2 | 62.5 | 71.1 | 64.4 | **81.6** | 70.4 |
| CLUE (Prabhu et al., 2021) | 57.6 | 77.5 | 58.6 | 58.9 | 76.8 | 65.9 | 66.3 | 60.2 | 60.5 | 66.2 | 58.7 | 76.0 | 65.3 |
| EADA (Xie et al., 2021) | 66.0 | 80.8 | 63.5 | 69.4 | 83.0 | 65.1 | 71.1 | 68.6 | 65.7 | 71.0 | 64.3 | 81.0 | 70.8 |
| **DUC** | 67.1±0.4 | 81.1±0.5 | **67.1±0.5** | **74.0±0.6** | 83.5±0.3 | **67.6±0.3** | **72.4±0.7** | **70.3±0.4** | **66.5±0.4** | **73.5±0.3** | **70.0±0.5** | 81.1±0.3 | **72.9±0.4** |
| Fully-supervised | 74.8 | 89.2 | 73.8 | 82.9 | 89.2 | 75.1 | 82.4 | 75.6 | 74.9 | 82.7 | 73.8 | 88.7 | 80.3 |

For miniDomainNet, since these compared baselines do not report the results on this dataset, we report our own runs based on their open source code.

Table 2: Accuracy (%) on Office-Home and VisDA-2017 with 5% target samples as the labeling budget (ResNet-50).

| Method | VisDA-2017 Synthetic→Real | Office-Home | | | | | | | | | | | | |
|---|---|---|---|---|---|---|---|---|---|---|---|---|---|---|
| | | Ar→Cl | Ar→Pr | Ar→Rw | Cl→Ar | Cl→Pr | Cl→Rw | Pr→Ar | Pr→Cl | Pr→Rw | Rw→Ar | Rw→Cl | Rw→Pr | Avg |
| Source-only | 44.7 ± 0.1 | 42.1 | 66.3 | 73.3 | 50.7 | 59.0 | 62.6 | 51.9 | 37.9 | 71.2 | 65.2 | 42.6 | 76.6 | 58.3 |
| Random | 78.1 ± 0.6 | 52.5 | 74.3 | 77.4 | 56.3 | 69.7 | 68.9 | 57.7 | 50.9 | 75.8 | 70.0 | 54.6 | 81.3 | 65.8 |
| BvSB (Joshi et al., 2009) | 81.3 ± 0.4 | 56.3 | 78.6 | 79.3 | 58.1 | 74.0 | 70.9 | 59.5 | 52.6 | 77.2 | 71.2 | 56.4 | 84.5 | 68.2 |
| Entropy (Wang & Shang, 2014) | 82.7 ± 0.3 | 58.0 | 78.4 | 79.1 | 60.5 | 73.0 | 72.6 | 60.4 | 54.2 | 77.9 | 71.3 | 58.0 | 83.6 | 68.9 |
| CoreSet (Sener & Savarese, 2018) | 81.9 ± 0.3 | 51.8 | 72.6 | 75.9 | 58.3 | 68.5 | 70.1 | 58.8 | 48.8 | 75.2 | 69.0 | 52.7 | 80.0 | 65.1 |
| WAAL (Shui et al., 2020) | 83.9 ± 0.4 | 55.7 | 77.1 | 79.3 | 61.1 | 74.7 | 72.6 | 60.1 | 52.1 | 78.1 | 70.1 | 56.6 | 82.5 | 68.3 |
| BADGE (Ash et al., 2020) | 84.3 ± 0.3 | 58.2 | 79.7 | 79.9 | 61.5 | 74.6 | 72.9 | 61.5 | 56.0 | 78.3 | 71.4 | 60.9 | 84.2 | 69.9 |
| AADA (Su et al., 2019) | 80.8 ± 0.4 | 56.6 | 78.1 | 79.0 | 58.5 | 73.7 | 71.0 | 60.1 | 53.1 | 77.0 | 70.6 | 57.0 | 84.5 | 68.3 |
| DBAL (Deheeger et al., 2021) | 82.6 ± 0.3 | 58.7 | 77.3 | 79.2 | 61.7 | 73.8 | 73.3 | 62.6 | 54.5 | 78.1 | 72.4 | 59.9 | 84.3 | 69.6 |
| TQS (Fu et al., 2021) | 83.1 ± 0.4 | 58.6 | 81.1 | 81.5 | 61.1 | 76.1 | 73.3 | 61.2 | 54.7 | 79.7 | 73.4 | 58.9 | 86.1 | 70.5 |
| CLUE (Prabhu et al., 2021) | 85.2 ± 0.4 | 58.0 | 79.3 | 80.9 | 68.8 | 77.5 | 76.7 | 66.3 | 57.9 | 81.4 | 75.6 | 60.8 | 86.3 | 72.5 |
| EADA (Xie et al., 2021) | 88.3 ± 0.1 | 63.6 | 84.4 | 83.5 | 70.7 | **83.7** | 80.5 | 73.0 | 63.5 | 85.2 | 78.4 | 65.4 | 88.6 | 76.7 |
| **DUC** | 88.9 ± 0.2 | **65.5±0.3** | **84.9±0.2** | **84.3±0.4** | **73.0±0.4** | 83.4±0.2 | **81.1±0.3** | **73.9±0.3** | **66.6±0.5** | **85.4±0.2** | **80.1±0.2** | **69.2±0.3** | **88.8±0.1** | **78.0±0.3** |
| Fully-supervised | 93.3 | 95.6 | 99.5 | 99.5 | 99.3 | 99.6 | 99.5 | 99.3 | 95.8 | 99.5 | 99.5 | 95.6 | 99.5 | 98.5 |

## 4 EXPERIMENTS

### 4.1 EXPERIMENTAL SETUP

We evaluate DUC on three cross-domain image classification datasets: *miniDomainNet* (Zhou et al., 2021), *Office-Home* (Venkateswara et al., 2017), *VisDA-2017* (Peng et al., 2017), and two adaptive semantic segmentation tasks: *GTAV (Richter et al., 2016) → Cityscapes (Cordts et al., 2016)*, *SYNTHIA (Ros et al., 2016) → Cityscapes*. For image classification, we use ResNet-50 (He et al., 2016) pre-trained on ImageNet (Deng et al., 2009) as the backbone. Following (Xie et al., 2021), the total labeling budget $B$ is set as $5\%$ of target samples, which is divided into 5 selection steps, i.e., the labeling budget in each selection step is $b = B/5 = 1\% \times n_t$. We adopt the mini-batch SGD optimizer with batch size 32, momentum 0.9 to optimize the model. As for hyper-parameters, we select them by the grid search with deep embedded validation (DEV) (You et al., 2019) and use $\beta = 1.0, \lambda = 0.05, \kappa = 10$ for image classification. For semantic segmentation, we adopt DeepLab-v2 (Chen et al., 2015) and DeepLab-v3+ (Chen et al., 2018) with the backbone ResNet-101 (He et al., 2016), and totally annotate $5\%$ pixels of target images. Similarly, the mini-batch SGD optimizer is adopted, where batch size is 2. And we set $\beta = 1.0, \lambda = 0.01, \kappa = 10$ for semantic segmentation. For all tasks, we report the mean±std of 3 random trials, and we perform fully supervised training with the labels of all target data as the upper bound. Detailed dataset description and implementation details are given in Sec C of the appendix. Code is available at `https://github.com/BIT-DA/DUC`.

### 4.2 MAIN RESULTS

#### 4.2.1 IMAGE CLASSIFICATION

**Results on miniDomainNet** are summarized in Table 1, where clustering-based methods (e.g., DBAL, CLUE) seem to be less effective than uncertainty-based methods (e.g., TQS, EADA) on the large-scale dataset. This may be because the clustering becomes more difficult with the increase of data scale. Contrastively, our method works well with the large-scale dataset. Moreover, DUC surpasses the most competitive rival EADA by 2.1% on average accuracy. This is owed to our better estimation of predictive uncertainty by interpreting the prediction as a distribution, while EADA only considers the point estimate of predictions, which can easily be miscalibrated.

**Results on Office-Home** are reported in Table 2, where active DA methods (e.g., DBAL, TQS, CLUE) generally outperform AL methods (e.g., Entropy, CoreSet, WAAL), showing the necessity of

Table 3: mIoU (%) comparisons on the task GTAV → Cityscapes.

| Method | budget | road | side. | buil. | wall | fence | pole | light | sign | veg. | terr. | sky | pers. | rider | car | truck | bus | train | motor | bike | mIoU |
|---|---|---|---|---|---|---|---|---|---|---|---|---|---|---|---|---|---|---|---|---|---|
| Source-only | - | 75.8 | 16.8 | 77.2 | 12.5 | 21.0 | 25.5 | 30.1 | 20.1 | 81.3 | 24.6 | 70.3 | 53.8 | 26.4 | 49.9 | 17.2 | 25.9 | 6.5 | 25.3 | 36.0 | 36.6 |
| MRKLD (Zou et al., 2019) | - | 91.0 | 55.4 | 80.0 | 33.7 | 21.4 | 37.3 | 32.9 | 24.5 | 85.0 | 34.1 | 80.8 | 57.7 | 24.6 | 84.1 | 27.8 | 30.1 | 26.9 | 26.0 | 42.3 | 47.1 |
| Seg-Uncertainty (Zheng & Yang, 2021) | - | 90.4 | 31.2 | 85.1 | 36.9 | 25.6 | 37.5 | 48.8 | 48.5 | 85.3 | 34.8 | 81.1 | 64.4 | 36.8 | 86.3 | 34.9 | 52.2 | 1.7 | 29.0 | 44.6 | 50.3 |
| TPLD (Shin et al., 2020) | - | 94.2 | 60.5 | 82.8 | 36.6 | 16.6 | 39.3 | 29.0 | 25.5 | 85.6 | 44.9 | 84.4 | 60.6 | 27.4 | 84.1 | 37.0 | 47.0 | 31.2 | 36.1 | 50.3 | 51.2 |
| ProDA (Zhang et al., 2021) | - | 87.8 | 56.0 | 79.7 | 46.3 | 44.8 | 45.6 | 53.5 | 53.5 | 88.6 | 45.2 | 82.1 | 70.7 | 39.2 | 88.8 | 45.5 | 59.4 | 1.0 | 48.9 | 56.4 | 57.5 |
| EADA (Xie et al., 2021) | 5% | - | - | - | - | - | - | - | - | - | - | - | - | - | - | - | - | - | - | - | 65.2 |
| EADA* (Xie et al., 2021) | 5% | 96.5 | 73.8 | 88.6 | 51.3 | 44.8 | 40.9 | 47.4 | 56.5 | 89.1 | 55.0 | 91.3 | 69.2 | 47.6 | 90.7 | 66.4 | 64.9 | 53.1 | 52.4 | 66.6 | 65.6 |
| **DUC** | 5% | **96.8** | **76.2** | 89.2 | **53.2** | 46.0 | 42.5 | 48.5 | 57.6 | 89.6 | **58.5** | 92.1 | 72.9 | 51.3 | 92.0 | 62.8 | 72.2 | 48.5 | 52.8 | 70.3 | 67.0 |
| Fully-supervised | 100% | 97.2 | 78.1 | 90.6 | 54.5 | 52.7 | 43.2 | 54.2 | 65.1 | 90.5 | 59.9 | 92.4 | 72.8 | 50.7 | 91.8 | 74.0 | 77.2 | 67.6 | 56.3 | 70.9 | 70.5 |
| AADA# (Su et al., 2019) | 5% | 92.2 | 59.9 | 87.3 | 36.4 | 45.7 | 46.1 | 50.6 | 59.5 | 88.3 | 44.0 | 90.2 | 69.7 | 38.2 | 90.0 | 55.3 | 45.1 | 32.0 | 32.6 | 62.9 | 59.3 |
| MADA# (Ning et al., 2021) | 5% | 95.1 | 69.8 | 88.5 | 43.3 | **48.7** | 45.7 | 53.3 | 59.2 | 89.1 | 46.7 | 91.5 | 73.9 | 50.1 | 91.2 | 60.6 | 56.9 | 48.4 | 51.6 | 68.7 | 64.9 |
| **DUC#** | 5% | 95.9 | 70.6 | **89.8** | 50.7 | 48.3 | **47.8** | **53.7** | **59.7** | **90.3** | 56.8 | **93.1** | **74.7** | **55.1** | **92.8** | **74.8** | **77.9** | 63.4 | **59.5** | **71.6** | **69.8** |
| Fully-supervised# | 100% | 96.8 | 80.4 | 90.2 | 48.6 | 56.8 | 52.3 | 58.6 | 68.3 | 90.2 | 59.4 | 93.3 | 75.8 | 54.2 | 92.5 | 74.9 | 79.1 | 71.6 | 56.8 | 71.8 | 72.2 |

Methods with # are based on DeepLab-v3+ (Chen et al., 2018) and others are based on DeepLab-v2 (Chen et al., 2015). Method with budget "-" are the source-only or UDA methods. EADA* denotes the results are based on our own runs according to the corresponding open source code.

Table 4: mIoU (%) comparisons on the task SYNTHIA → Cityscapes. mIoU* is reported according to the average of 13 classes, excluding the "wall", "fence" and "pole".

| Method | budget | road | side. | buil. | wall* | fence* | pole* | light | sign | veg. | sky | pers. | rider | car | bus | motor | bike | mIoU | mIoU* |
|---|---|---|---|---|---|---|---|---|---|---|---|---|---|---|---|---|---|---|---|
| Source-only | - | 64.3 | 21.3 | 73.1 | 2.4 | 1.1 | 31.4 | 7.0 | 27.7 | 63.1 | 67.6 | 42.2 | 19.9 | 73.1 | 15.3 | 10.5 | 38.9 | 34.9 | 40.3 |
| MRKLD (Zou et al., 2019) | - | 67.7 | 32.2 | 73.9 | 10.7 | 1.6 | 37.4 | 22.2 | 31.2 | 80.8 | 80.5 | 60.8 | 29.1 | 82.8 | 25.0 | 19.4 | 45.3 | 43.8 | 50.1 |
| TPLD (Shin et al., 2020) | - | 80.9 | 44.3 | 82.2 | 19.9 | 0.3 | 40.6 | 20.5 | 30.1 | 77.2 | 80.9 | 60.6 | 25.5 | 84.8 | 41.1 | 24.7 | 43.7 | 47.3 | 53.5 |
| Seg-Uncertainty (Zheng & Yang, 2021) | - | 87.6 | 41.9 | 83.1 | 14.7 | 1.7 | 36.2 | 31.3 | 19.9 | 81.6 | 80.6 | 63.0 | 21.8 | 86.2 | 40.7 | 23.6 | 53.1 | 47.9 | 54.9 |
| ProDA (Zhang et al., 2021) | - | 87.8 | 45.7 | 84.6 | 37.1 | 0.6 | 44.0 | 54.6 | 37.0 | 88.1 | 84.4 | 74.2 | 24.3 | 88.2 | 51.1 | 40.5 | 45.6 | 55.5 | 62.0 |
| **DUC** | 5% | 96.1 | 73.1 | 88.7 | 43.3 | 39.0 | 42.2 | 49.9 | 55.5 | **90.7** | **92.8** | 73.7 | 49.2 | 91.9 | 67.9 | 45.9 | **71.1** | 66.9 | 72.8 |
| Fully-supervised | 100% | 97.3 | 79.4 | 89.6 | 52.8 | 54.0 | 46.7 | 53.4 | 62.6 | 90.5 | 92.9 | 71.3 | 50.8 | 92.1 | 77.9 | 55.4 | 68.7 | 71.0 | 75.5 |
| AADA# (Su et al., 2019) | 5% | 91.3 | 57.6 | 86.9 | 37.6 | **48.3** | 45.0 | 50.4 | 58.5 | 88.2 | 90.3 | 69.4 | 37.9 | 89.9 | 44.5 | 32.8 | 62.5 | 61.9 | 66.2 |
| MADA# (Ning et al., 2021) | 5% | **96.5** | **74.6** | 88.8 | 45.9 | 43.8 | 46.7 | 52.4 | 60.5 | 89.7 | 92.2 | 74.1 | 51.2 | 90.9 | 60.3 | 52.4 | 69.4 | 68.1 | 73.3 |
| **DUC#** | 5% | 96.3 | **74.6** | **89.4** | **46.8** | 47.6 | **46.8** | 49.7 | **63.1** | 90.3 | 91.3 | **74.7** | **53.8** | **93.1** | **78.9** | **57.0** | 71.0 | **70.3** | **75.6** |
| Fully-supervised# | 100% | 97.0 | 80.4 | 90.9 | 48.6 | 56.2 | 52.1 | 58.5 | 67.4 | 91.3 | 93.4 | 75.5 | 54.2 | 92.3 | 78.5 | 56.1 | 71.3 | 72.7 | 77.4 |

Methods with # are based on DeepLab-v3+ (Chen et al., 2018) and others are based on DeepLab-v2 (Chen et al., 2015). Method with budget "-" are the source-only or UDA methods.

considering targetness. And our method beats EADA by $1.3\%$, validating the efficacy of regrading the prediction as a distribution and selecting data based on both the distribution and data uncertainties.

**Results on VisDA-2017** are given in Table 2. On this large-scale dataset, our method still works well, achieving the highest accuracy of $88.9\%$, which further validates the effectiveness of our approach.

### 4.2.2 SEMANTIC SEGMENTATION

**Results on GTAV→Cityscapes** are shown in Table 3. Firstly, we can see that with only 5% labeling budget, the performance of domain adaptation can be significantly boosted, compared with UDA methods. Besides, compared with active DA methods (AADA and MADA), our DUC largely surpasses them according to mIoU: DUC (69.8, **10.5↑**) v.s. AADA (59.3), DUC (69.8, **4.9↑**) v.s. MADA (64.9). This can be explained as the more informative target samples selected by DUC and the implicitly mitigated domain shift by reducing the distribution uncertainty of unlabeled target data.

**Results on SYNTHIA→Cityscapes** are presented in Table 4. Due to the domain shift from virtual to realistic as well as a variety of driving scenes and weather conditions, this adaptation task is challenging, while our method still achieves considerable improvements. Concretely, according to the average mIoU of 16 classes, DUC exceeds AADA and MADA by 8.4% and 2.2%, respectively. We owe the advances to the better measurement of targetness and discriminability, which are both closely related with the prediction. Thus the selected target pixels are really conducive to the classification.

### 4.3 ANALYTICAL EXPERIMENTS

**Ablation Study.** Firstly, we try EDL with different active selection criteria. According to the results of the first four rows in Table 5, variant A obviously surpasses EDL with other selection criteria and demonstrates the superiority of evaluating the sample value from multiple aspects, where variant A is actually equivalent to EDL with our two-round selection strategy. Then, we study the effects of $\mathcal{L}_{U_{dis}}$ and $\mathcal{L}_{U_{data}}$. The superiority of variant B over A manifests the necessity of reducing the distributional uncertainty of unla-

Table 5: Ablation study of DUC on Office-Home.

| Method | Loss | | Active Selection Criterion | | | | Avg |
|---|---|---|---|---|---|---|---|
| | $\mathcal{L}_{U_{dis}}$ | $\mathcal{L}_{U_{data}}$ | random | entropy | $U_{dis}$ | $U_{data}$ | |
| EDL | - | - | - | - | - | - | 61.5 |
| EDL | - | - | ✓ | - | - | - | 71.1 |
| EDL | - | - | - | ✓ | - | - | 73.3 |
| Variant A | - | - | - | - | ✓ | ✓ | 74.1 |
| Variant B | ✓ | - | - | - | ✓ | ✓ | 76.6 |
| Variant C | - | ✓ | - | - | ✓ | ✓ | 74.2 |
| Variant D | ✓ | ✓ | - | - | - | - | 68.6 |
| Variant E | ✓ | ✓ | ✓ | - | - | - | 75.0 |
| Variant F | ✓ | ✓ | - | ✓ | - | - | 76.7 |
| Variant G | ✓ | ✓ | - | - | ✓ | - | 77.1 |
| Variant H | ✓ | ✓ | - | - | - | ✓ | 76.9 |
| DUC | ✓ | ✓ | - | - | ✓ | ✓ | **78.0** |

beled samples, by which domain shift is mitigated. Another interesting observation is that $\mathcal{L}_{U_{data}}$ does not bring obvious boosts to variant A. We infer this is because the distribution uncertainty will potentially affect the data uncertainty, as shown in Eq. (2). It is meaningless to reduce $U_{data}$

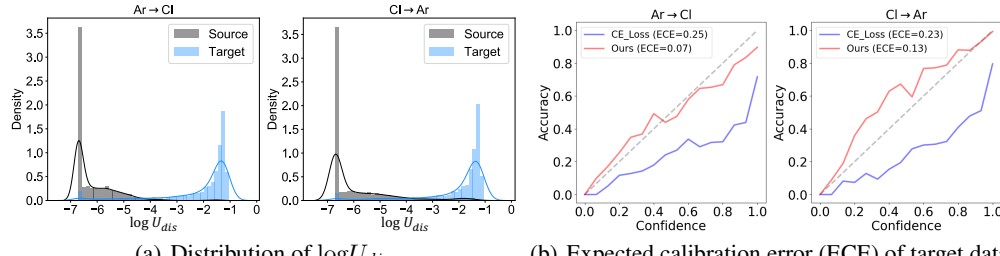

(a) Distribution of $\log U_{dis}$        (b) Expected calibration error (ECE) of target data

Figure 3: (a): The distribution of $\log U_{dis}$ for source and target data on task $Ar \rightarrow Cl$ and $Cl \rightarrow Ar$. For elegancy, we apply logarithm to $U_{dis}$. (b): Expected calibration error (ECE) of target data, where the standard DNN with cross entropy (CE) loss and our model are both trained with source data.

when $U_{dis}$ is large, because opinions are derived from insufficient evidences and unreliable. Instead, reducing both $U_{dis}$ and $U_{data}$ is the right choice, which is further verified by the $7.1\%$ improvements of variant D over pure EDL. Besides. Even without $\mathcal{L}_{U_{dis}}$ and $\mathcal{L}_{U_{data}}$, variant A still exceeds CLUE by $1.6\%$, showing our superiority. Finally, we try different selection strategies. Variant E to H denote only one criterion is used in the selection. We see DUC beats variant F, G, H, since the entropy degenerates into the uncertainty based on point estimate, while $U_{dis}$ or $U_{data}$ only considers either targetness or discriminability. Contrastively, DUC selects samples with both characteristics.

**Distribution of $U_{dis}$ Across Domains.** To answer whether the distribution uncertainty can represent targetness, we plot in Fig. 3(a) the distribution of $U_{dis}$, where the model is trained on source domain with $\mathcal{L}_{edl}$. We see that the $U_{dis}$ of target data is noticeably biased from source domain. Such results show that our $U_{dis}$ can play the role of domain discriminator without introducing it. Moreover, the score of domain discriminator is not directly linked with the prediction, which causes the selected samples not necessarily beneficial for classifier, while our $U_{dis}$ is closely related with the prediction.

**Expected Calibration Error (ECE).** Following (Joo et al., 2020), we plot the expected calibration error (ECE) (Naeini et al., 2015) on target data in Fig. 3(b) to evaluate the calibration. Obviously, our model presents better calibration performance, with much lower ECE. While the accuracy of standard DNN is much lower than the confidence, when the confidence is high. This implies that standard DNN can easily produce overconfident but wrong predictions for target data, leading to the estimated predictive uncertainty unreliable. Contrastively, DUC mitigates the miscalibration problem.

**Effect of Selection Ratio in the First Round.** Hyper-parameter $\kappa$ controls the selection ratio in the first round and Fig. 4(a) presents the results on Office-Home with different $\kappa$. The performance with too much or too small $\kappa$ is inferior, which results from the imbalance between targetness and discriminability. When $\kappa = 1$ or $\kappa = 100$, the selection degenerates to the one-round sampling manner according to $U_{dis}$ and $U_{data}$, respectively. In general, we find $\kappa \in \{10, 20, 30\}$ works better.

**Hyper-parameter Sensitivity.** $\beta$ and $\lambda$ control the tradeoff between $\mathcal{L}_{U_{dis}}$ and $\mathcal{L}_{U_{data}}$. we test the sensitivity of the two hyper-parameters on the Office-Home dataset. The results are presented in Fig. 4(b), where $\beta \in \{0.01, 0.05, 0.1, 0.5, 1.0\}$ and $\lambda \in \{0.001, 0.005, 0.01, 0.05, 0.1\}$. According to the results, DUC is not that sensitive to $\beta$ but is a little bit sensitive to $\lambda$. In general, we recommend $\lambda \in \{0.01, 0.05, 0.1\}$ for trying.

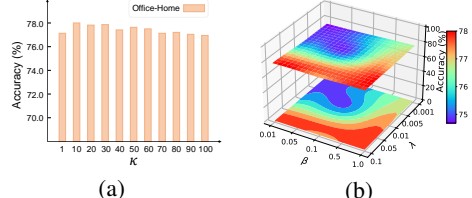

(a)        (b)

Figure 4: (a): Effect of different first-round selection ration $\kappa\%$ on Office-Home. (b): Hyper-parameter sensitivity of $\beta, \lambda$ on Office-Home.

## 5   CONCLUSION

In this paper, we address active domain adaptation (DA) from the evidential perspective and propose a Dirichlet-based Uncertainty Calibration (DUC) approach. Compared with existing active DA methods which estimate predictive uncertainty based on the the prediction of deterministic models, we interpret the prediction as a distribution on the probability simplex via placing a Dirichlet prior on the class probabilities. Then, based on the prediction distribution, two uncertainties from different origins are designed in a unified framework to select informative target samples. Extensive experiments on both image classification and semantic segmentation verify the efficacy of DUC.

ACKNOWLEDGEMENTS

This work was supported by National Key R&D Program of China (No. 2021YFB3301503).

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

APPENDIX CONTENTS

## A  BROADER IMPACT AND LIMITATIONS

Our work focuses on active domain adaptation (DA), which aims to maximally improve the model adaptation from one labeled domain (termed source domain) to another unlabeled domain (termed target domain) by annotating limited target data. In this paper, we suggest a new perspective for active DA and further boost the adaptation performances on both cross-domain image classification and semantic segmentation benchmarks. The advances mean that our method may potentially benefit relevant social activities, e.g., commodity classification, autonomous driving in different scenes, without consuming high labor cost to annotate massive new data for different scenes. While we do not anticipate adverse impacts, our method may suffer from some limitations. For example, our work is restricted to classification and segmentation tasks in this paper. In the future, we will explore our method in other tasks, e.g., object detection and regression, hoping to benefit more diverse fields. Besides, we only try to train the Dirichlet-based model using the evidential deep learning in the paper. Yet, there may exist better training frameworks, e.g., normalizing flow-based Dirichlet Posterior Network which can predict a closed-form posterior distribution over predicted probabilities for any input sample. In the future, we may also explore to extend our approach into the training framework of normalizing flow-based Dirichlet Posterior Network.

## B  ALGORITHM OF DUC

The training procedure of DUC is shown in Algorithm 1.

---

**Algorithm 1** Pseudo code of the proposed DUC

---

**Input:** labeled source dataset $\mathcal{S}$, unlabeled target dataset $\mathcal{T}$, selection steps $R$, total annotation budget $B$, hyperparameters $\kappa, \beta, \lambda$, total training steps $T$.
**Output:** learned model parameters $\boldsymbol{\theta}$.
1: Initialize model parameters $\boldsymbol{\theta}$.
2: Define $\mathcal{T}^l = \varnothing$ and $\mathcal{T}^u = \mathcal{T}$, $b = \frac{B}{|R|}$.
3: **for** $t = 1$ **to** $T$ **do**
4:     Update parameters $\boldsymbol{\theta}$ via minimizing $\mathcal{L}_{total}$.
5:     **if** $t \in R$ **then**
6:         $\forall \boldsymbol{x}_j \in \mathcal{T}^u$, compute its distribution and data uncertainties: $U_{dis}(\boldsymbol{x}_j, \boldsymbol{\theta}), U_{data}(\boldsymbol{x}_j, \boldsymbol{\theta})$.
7:         $temp\_Candi \leftarrow$ select top $\kappa b$ samples with highest $U_{dis}$ from $\mathcal{T}^u$.
8:         $Candi \leftarrow$ select top $b$ samples with highest $U_{data}$ from $temp\_Candi$.
9:         Query the labels of $Candi$ from the oracle.
10:         $\mathcal{T}^u = \mathcal{T}^u \backslash Candi, \mathcal{T}^l = \mathcal{T}^l \cup Candi$.
11:     **end if**
12: **end for**
13: **return** Final model parameters $\boldsymbol{\theta}$.

---

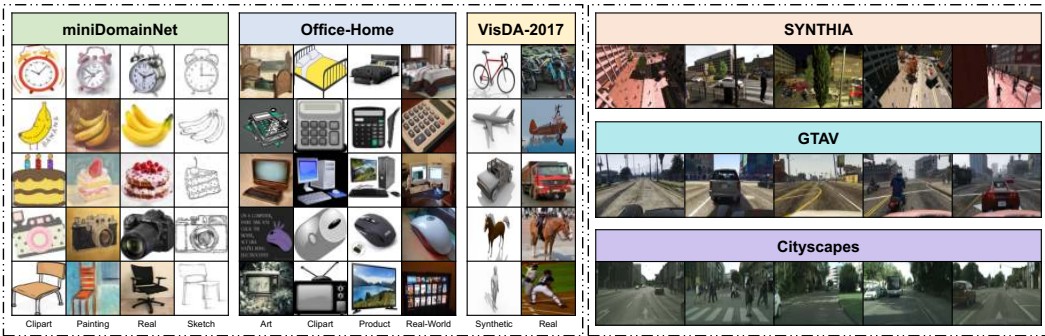

Figure 5: Image examples from dataset miniDomainNet, Office-Home, VisDA-2017, Cityscapes, GTAV and SYNTHIA.

## C EXPERIMENTAL SETUP DETAILS

### C.1 DATASET DESCRIPTION

**miniDomainNet** (Zhou et al., 2021) is a subset of DomainNet (Peng et al., 2019), a large-scale image classification dataset for domain adaptation. miniDomainNet contains more than 130,000 images of 126 classes from four domains: Clipart (clp), Painting (pnt), Real (rel) and Sketch (skt). The large data scale and multiplicity make the adaptation on this dataset quite challenging. And we build 12 adaptation tasks: clp→pnt, · · · , skt→rel, by permuting the four domains, to evaluate our method.

**Office-Home** (Venkateswara et al., 2017) collects 15,500 images of 65 categories from office and home scenes. And these images are divided into four distinct domains: Art (Ar), Clipart (Cl), Product (Pr) and Real-World (Rw), respectively with images from artistic depictions, clipart pictures, product pictures and cameras.

**VisDA-2017** (Peng et al., 2017) is a large scale dataset for cross-domain image classification. It collects images of 12 classes, including synthetic images rendered from 3D models and real images. Following Xie et al. (2021), we use 152,397 synthetic images as source domain and 72,372 real images as target domain, forming the adaptation task: Synthetic→Real.

**Cityscapes** (Cordts et al., 2016) gathers 5,000 images of urban street scenes from real world, where each pixel in the image is annotated from 19 categories and the image resolution is 2048×1024. These images are divided into training, validation and test splits. Similar to (Ning et al., 2021), we use the training split with 2,975 images as target training data, where labels are not used, and the model is evaluated on the validation split with 500 images by reporting the mIoU of the common categories.

**GTAV** (Richter et al., 2016) is a dataset of 24,966 simulated images with pixel level semantic annotation. These images are rendered by "Grand Theft Auto V" game engine, with the resolution of 1914×1052. And this dataset shares 19 categories with the Cityscapes dataset.

**SYNTHIA** (Ros et al., 2016) consists of 9,400 synthetic images of street scenes, with the image resolution of 1280×760. It contains diverse street scenes, such as towns and highways, different weather conditions and seasons. There are 16 categories that are compatible with the semantic categories in Cityscapes.

The image illustration of different datasets is shown in Fig. 5.

### C.2 IMPLEMENTATION DETAILS

IMAGE CLASSIFICATION

All experiments are implemented via PyTorch (Paszke et al., 2019). For image classification, we use ResNet-50 (He et al., 2016) pre-trained on ImageNet (Deng et al., 2009) as the backbone, and the exponential function is employed to the model output to ensure $\alpha$ non-negative. Following (Xie et al., 2021; Fu et al., 2021), The total labeling budget $B$ is set as $5\%$ of target samples, which is

Table 6: Results with different total labeling budget $B$ on Office-Home (ResNet-50).

| Total labeling budget $B$ | 0% | 2.5% | 5% | 7.5% | 10% | 12.5% | 15% | 17.5% | 20% |
|---|---|---|---|---|---|---|---|---|---|
| Avg accuracy | 68.6 | 72.9 | 78.0 | 80.2 | 82.4 | 84.8 | 86.4 | 88.0 | 88.9 |
| Gain over previous one | - | 4.3↑ | 5.1↑ | 2.2↑ | 2.2↑ | 2.4↑ | 1.6↑ | 1.6↑ | 0.9↑ |

Table 7: Accuracy (%) on Office-Home with 5% target samples as the labeling budget (ResNet-50), when DUC is combined with semi-supervised learning method.

| Method | Ar→Cl | Ar→Pr | Ar→Rw | Cl→Ar | Cl→Pr | Cl→Rw | Pr→Ar | Pr→Cl | Pr→Rw | Rw→Ar | Rw→Cl | Rw→Pr | Avg |
|---|---|---|---|---|---|---|---|---|---|---|---|---|---|
| **DUC** | 65.5 | 84.9 | 84.3 | 73.0 | 83.4 | 81.1 | 73.9 | 66.6 | 85.4 | 80.1 | 69.2 | 88.8 | 78.0 |
| DUC w/ $\mathcal{L}_{nll}^{fixmatch}$ | 66.5 | 85.7 | 85.0 | 73.3 | 84.3 | 82.8 | 74.8 | 67.0 | 85.7 | 81.5 | 70.8 | 89.7 | 78.9 |

divided into 5 selection steps, i.e., the labeling budget in each selection step is $b = B/5 = 1\% \times n_t$. For data preprocessing, we use RandomHorizontalFlip, RandomResizedCrop and ColorJitter during the training process and use CenterCrop during the test stage. For the optimizer, we adopt the mini-batch stochastic gradient descent (SGD) optimizer with batch size 32, momentum 0.9, weight decay 0.001 and the learning rate schedule strategy in (Long et al., 2018). The initial learning rates for miniDomainNet, Office-Home and VisDA-2017 are 0.002, 0.004 and 0.001, respectively. As for hyper-parameters, we select them by the grid search and finally use $\beta = 1.0, \lambda = 0.05, \kappa = 10$ for miniDomainNet and Office-Home datasets. We run each task on a single NVIDIA GeForce RTX 2080 Ti GPU.

SEMANTIC SEGMENTATION

For semantic segmentation, we also implements the experiment using PyTorch (Paszke et al., 2019) and adopt the DeepLab-v2 (Chen et al., 2015) and DeepLab-v3+ (Chen et al., 2018) with the backbone ResNet-101 (He et al., 2016) pre-trained on ImageNet (Deng et al., 2009). Regarding the total labeling budget $B$, we totally annotate 5% pixels of target images, which is divided into 5 steps. In other words, we annotate 1% pixels for every image in each active selection step. For data preprocessing, source images are resized into 1280×720 and target images are resized into 1280×640. Similarly, the model is optimized using the mini-batch SGD optimizer with batch size 2, momentum 0.9, weight decay 0.0005. The "poly" learning rate schedule strategy with initial learning rate of 3e-4 is employed. And we set $\beta = 1.0, \lambda = 0.01, \kappa = 10$ for the semantic segmentation tasks. For each semantic segmentation task, we run the experiment on a single NVIDIA GeForce RTX 3090 GPU.

# D  ADDITIONAL RESULTS

## D.1  EFFECTS OF THE ORDERING OF $U_{dis}, U_{data}$ IN TWO-ROUND SAMPLING

Since our selection strategy is a two-round sampling manner, there naturally exists the ordering of $U_{dis}$ and $U_{data}$ in the two rounds. In Table 8, we explore the influence of different orderings, where $U_{dis}, U_{data}$ denotes $U_{dis}$ and $U_{data}$ are respectively used in the first and second round. We notice that $U_{dis}, U_{data}$ generally surpasses $U_{data}, U_{dis}$. It shows that selecting discriminability-conducive samples from target-representative samples is better for active DA than the converse manner. Thus, we adopt $U_{dis}, U_{data}$ throughout the paper.

Table 8: Analysis on the ordering of $U_{dis}, U_{data}$.

| Dataset / Ordering | miniDomainNet (Avg) | Office-Home (Avg) |
|---|---|---|
| $U_{dis}, U_{data}$ | 72.9 | 78.0 |
| $U_{data}, U_{dis}$ | 72.2 | 77.5 |

| Dataset / Ordering | GTAV→Cityscapes (mIoU) | SYNTHIA→Cityscapes (mIoU / mIoU*) |
|---|---|---|
| $U_{dis}, U_{data}$ | 67.0 | 66.9 / 72.8 |
| $U_{data}, U_{dis}$ | 66.2 | 65.3 / 71.6 |

## D.2  PERFORMANCE GAIN WITH DIFFERENT LABELING BUDGETS

In Table 6, we present the performances on Office-Home dataset with different total labeling budget $B$. As expected, better performances can be obtained with more labeled target samples accessible.

In addition, we observe that the increasing speed of performance generally gets slower, as the total labeling budget increases. This observation demonstrates that all samples are not equally informative and our method can successfully select relatively informative samples. For example, when labeling budget increasing from 17.5% to 20%, the performance gain is much smaller, which implies that the majority of informative samples has been selected by our method.

### D.3 Combination with Semi-supervised Learning

To further improve the performance, one can incorporate ideas from semi-supervised learning to use the unlabeled target data in training as well. Here, we consider one representative semi-supervised learning method: FixMatch (Sohn et al., 2020). Specifically, we apply strong and weak augmentations to each unlabeled target sample $x_j$, obtaining two views $x_j^{strong}$ and $x_j^{weak}$. And we use the pseudo label of weakly augmented view $x_j^{weak}$ as the label of strongly augmented view $x_j^{strong}$. Then the model is trained to minimize the loss $\mathcal{L}_{nll}^{fixmatch}$, i.e., the negative logarithm of the marginal likelihood of strongly augmented views. Concretely, $\mathcal{L}_{nll}^{fixmatch}$ is formulated as

$$
\mathcal{L}_{nll}^{fixmatch} = \frac{1}{M} \sum_{x_j \in \mathcal{T}^u \wedge \tau < \max_c \bar{\rho}_{jc}^{weak}} -\log \left( \int p(y = \hat{y}_j^{weak} | \boldsymbol{\rho}) p(\boldsymbol{\rho} | x_j^{strong}, \boldsymbol{\theta}) d\boldsymbol{\rho} \right)
$$

$$
= \frac{1}{M} \sum_{x_j \in \mathcal{T}^u \wedge \tau < \max_c \bar{\rho}_{jc}^{weak}} \sum_{c=1}^{C} \hat{\Upsilon}_{jc}^{weak} \left( \log \left( \sum_{c=1}^{C} \alpha_{jc}^{strong} \right) - \log \alpha_{jc}^{strong} \right), \quad (9)
$$

where $\hat{y}_j^{weak} = \arg\max_c \bar{\rho}_{jc}^{weak} = \arg\max_c \mathbb{E}[Dir(\rho_c | \boldsymbol{\alpha}_j^{weak})]$ and $M = |\{x_j | x_j \in \mathcal{T}^u \wedge \tau < \max_c \bar{\rho}_{jc}^{weak}\}|$. $\tau$ is a hyper-parameter denoting the threshold above which the pseudo label is retained, and $\hat{\Upsilon}_{jc}^{weak}$ is the $c$-th element of the one-hot label vector $\hat{\boldsymbol{\Upsilon}}_j^{weak}$ for pseudo label $\hat{y}_j^{weak}$.

Table 7 presents the results on the Office-Home dataset when combining our method DUC with the semi-supervised learning method FixMatch Sohn et al. (2020), where the hyper-parameter $\tau$ is set to 0.8. We can see that utilizing unlabeled target data indeed conduces to improving the performance. Of course, other semi-supervised learning methods are also possible.

### D.4 Qualitative Visualization of Selected Samples

In the label histogram of Fig. 6, we plot the ground truth label distribution of the samples that are selected by DUC, with the total labeling budget $B = 5\% \times n_t$. For the Ar→Cl task, "Bottle", "Knives" and "Toys" are the top 3 classes that are picked, while "Bucket", "Pencil" and "Spoon" turn out to be the top 3 picked classes in the Cl→Ar task. It shows that our method DUC can adaptively select informative samples for different target domains. Despite few categories are not picked, we can still see that the samples selected by DUC are generally category-diverse. And, according to the visualization of selected samples, the style of target domain is indeed reflected in theses selected samples. In addition, we also visualize the selected pixels for the task GTAV→Cityscapes in Fig. 7. Overall, the selected pixels are from diverse objects that are hard to classify or are nearby together. Annotating such pixels can bring more beneficial knowledge for the model.

### D.5 t-SNE Visualization for Showing Effects of $\mathcal{L}_{U_{dis}}$

To verify that reducing our distribution uncertainty $U_{dis}$ conduces to the domain alignment, we respectively train the model with $\mathcal{L}_{edl}$ and $\mathcal{L}_{edl} + \beta \mathcal{L}_{U_{dis}}$, where there is no labeling budget. And the t-SNE (van der Maaten & Hinton, 2008) visualization of features from source and target domains on task Ar → Cl and Cl → Ar is shown in Fig. 8. Form the results, we can see that reducing the distribution uncertainty of target data indeed helps to alleviate the domain shift, which makes our method more suitable for active DA, compared with EDL Sensoy et al. (2018). Besides, the results also verify that our distribution uncertainty can measure the targetness of samples.

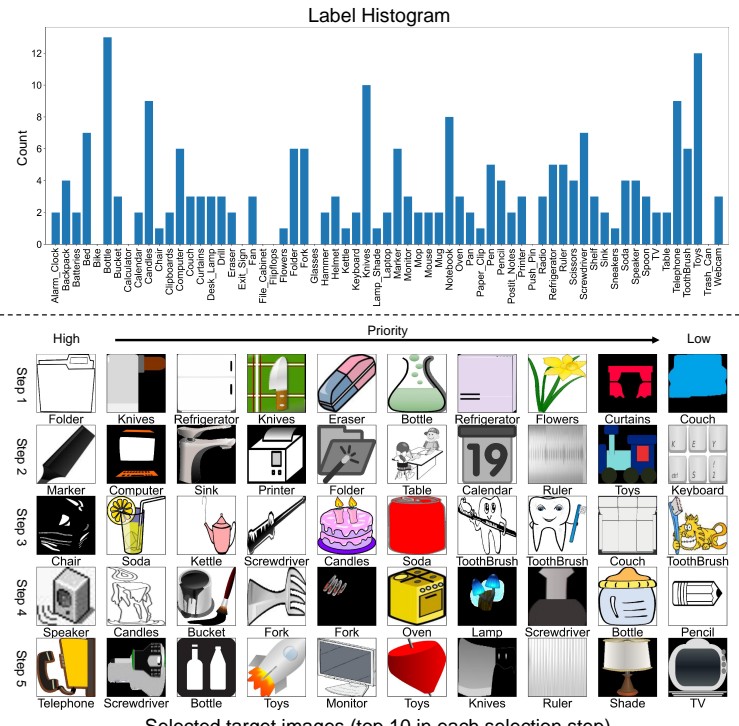

(a) Ar→Cl

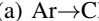

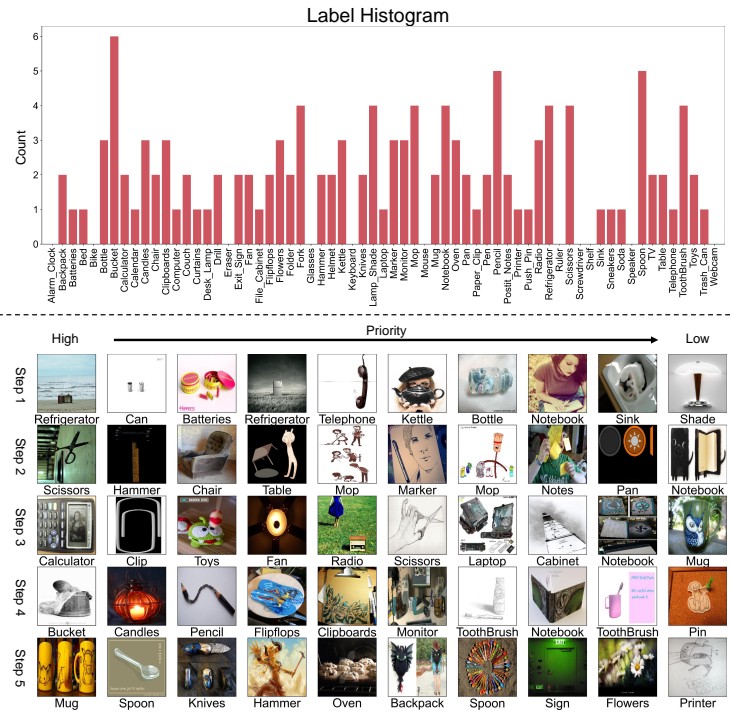

(b) Cl→Ar

Figure 6: (a) and (b) are the label histogram and examples of selected instances by DUC on Ar→Cl and Cl→Ar tasks, respectively. The total labeling budget is 5% of target images. For the visualization of selected samples, we present the top 10 selected instances in each active selection step.

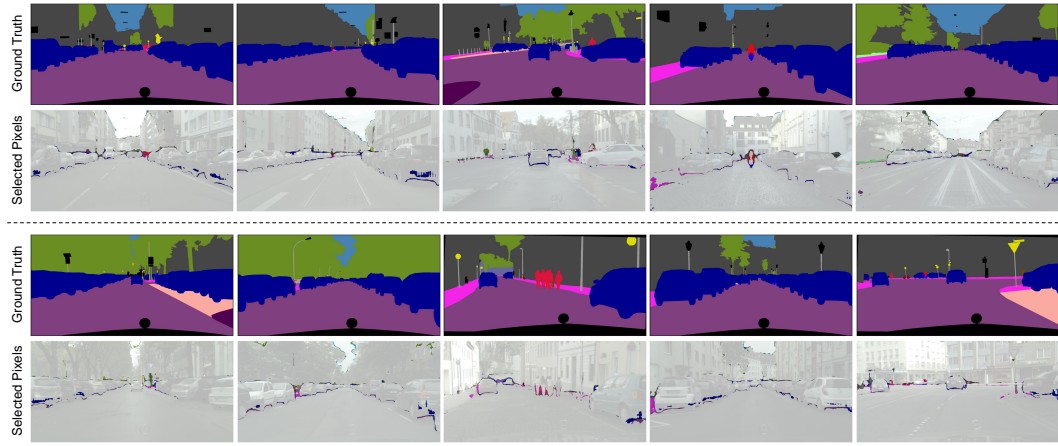

Figure 7: Visualization of selected pixels in the task GTAV→Cityscapes, with the total labeling budget of 5% pixels. Here, we randomly choose ten images from Cityscapes for display.

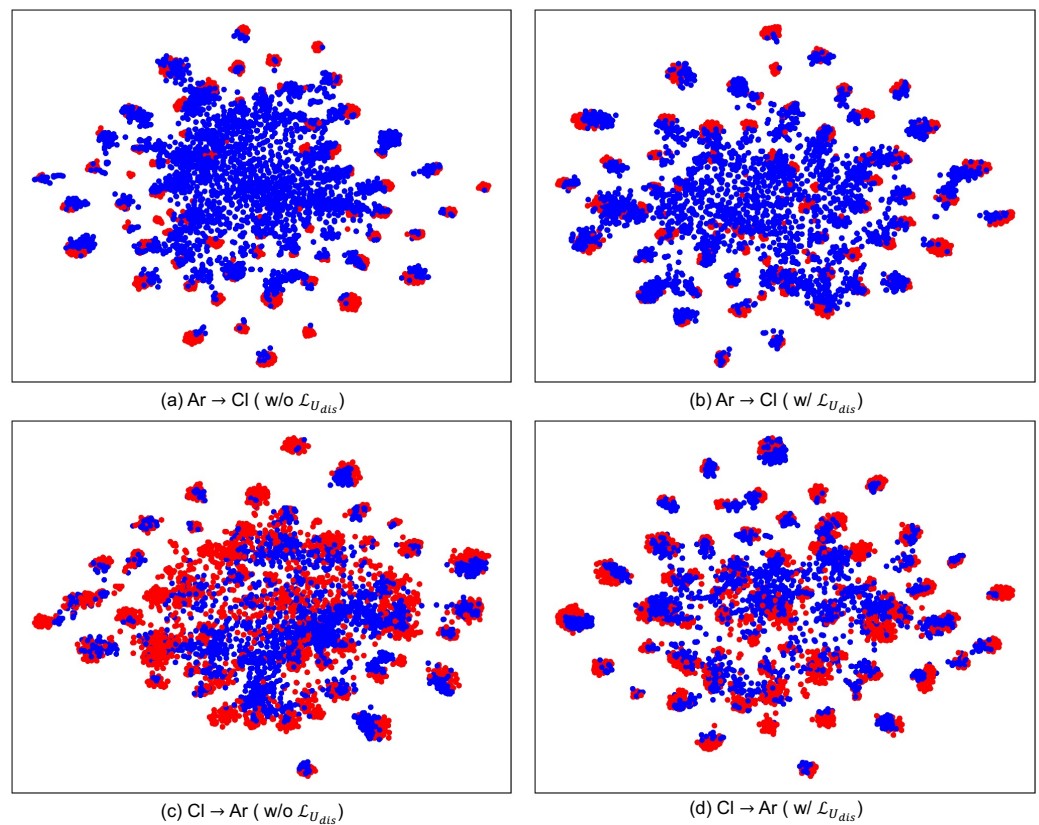

(a) Ar → Cl ( w/o $\mathcal{L}_{U_{dis}}$)

(b) Ar → Cl ( w/ $\mathcal{L}_{U_{dis}}$)

(c) Cl → Ar ( w/o $\mathcal{L}_{U_{dis}}$)

(d) Cl → Ar ( w/ $\mathcal{L}_{U_{dis}}$)

Figure 8: The t-SNE visualization of features learned by the model trained with $\mathcal{L}_{edl}$ and $\mathcal{L}_{edl} + \beta\mathcal{L}_{U_{dis}}$ respectively. Red and blue dots represent source and target features, respectively.

# E  DERIVATIONS

## E.1  PREDICTIVE PROBABILITY $P(y = c|\boldsymbol{x}_i, \boldsymbol{\theta})$

Given sample $\boldsymbol{x}_i$ and model $f$ parameterized with $\boldsymbol{\theta}$, the predicted class probability for class $c$ can be obtained as

$$
\begin{aligned}
P(y = c|\boldsymbol{x}_i, \boldsymbol{\theta}) &= \int p(y = c|\boldsymbol{\rho})p(\boldsymbol{\rho}|\boldsymbol{x}_i, \boldsymbol{\theta})d\boldsymbol{\rho} \\
&= \int \rho_c \cdot p(\boldsymbol{\rho}|\boldsymbol{x}_i, \boldsymbol{\theta})d\boldsymbol{\rho} \\
&= \int \int \cdots \int \rho_c \cdot p(\rho_1, \rho_2, \cdots, \rho_C|\boldsymbol{x}_i, \boldsymbol{\theta})d\rho_1 d\rho_2 \cdots d\rho_C \\
&= \int \rho_c \Big( \int \int \cdots \int \int \cdots \int p(\rho_1, \rho_2, \cdots, \rho_C|\boldsymbol{x}_i, \boldsymbol{\theta})d\rho_1 d\rho_2 \cdots d\rho_{c-1}d\rho_{c+1} \cdots d\rho_C \Big)d\rho_c \\
&= \int \rho_c \cdot p(\rho_c|\boldsymbol{x}_i, \boldsymbol{\theta})d\rho_c ,
\end{aligned}
\tag{10}
$$

where $\rho_c$ is the $c$-th element of the class probability vector $\boldsymbol{\rho}$. According to (Ng et al., 2011), the marginal distributions of Dirichlet is Beta distributions. Thus, given $p(\boldsymbol{\rho}|\boldsymbol{x}_i, \boldsymbol{\theta}) \sim Dir(\boldsymbol{\rho}|\boldsymbol{\alpha}_i)$, we have $p(\rho_c|\boldsymbol{x}_i, \boldsymbol{\theta}) \sim Beta(\rho_c|\alpha_{ic}, \alpha_{i0} - \alpha_{ic})$, where $\boldsymbol{\alpha}_i = g(f(\boldsymbol{x}_i, \boldsymbol{\theta}))$, $\alpha_{i0} = \sum_{k=1}^{C} \alpha_{ik}$ and $g(\cdot)$ is a function (e.g., exponential function) to keep $\boldsymbol{\alpha}_i$ (i.e., the parameters of Dirichlet distribution for sample $\boldsymbol{x}_i$) non-negative. And according to the probability density function of Beta distribution, we further have

$$
p(\rho_c|\boldsymbol{x}_i, \boldsymbol{\theta}) = \frac{1}{\mathcal{B}(\alpha_{ic}, \alpha_{i0} - \alpha_{ic})} \rho_c^{\alpha_{ic}-1}(1 - \rho_c)^{\alpha_{i0}-\alpha_{ic}-1},
\tag{11}
$$

where $\mathcal{B}(\cdot, \cdot)$ is the Beta function and $\mathcal{B}(\alpha_{ic}, \alpha_{i0} - \alpha_{ic}) = \frac{\Gamma(\alpha_{ic})\Gamma(\alpha_{i0}-\alpha_{ic})}{\Gamma(\alpha_{ic}+\alpha_{i0}-\alpha_{ic})}$, with $\Gamma(\cdot)$ denoting the Gamma function. Based on these, we can further derive $P(y = c|\boldsymbol{x}_i, \boldsymbol{\theta})$ as follows:

$$
\begin{aligned}
P(y = c|\boldsymbol{x}_i, \boldsymbol{\theta}) &= \int \rho_c \cdot p(\rho_c|\boldsymbol{x}_i, \boldsymbol{\theta})d\rho_c \\
&= \int \rho_c \cdot \Big( \frac{1}{\mathcal{B}(\alpha_{ic}, \alpha_{i0} - \alpha_{ic})} \rho_c^{\alpha_{ic}-1}(1 - \rho_c)^{\alpha_{i0}-\alpha_{ic}-1} \Big) d\rho_c \\
&= \frac{\mathcal{B}(\alpha_{ic} + 1, \alpha_{i0} - \alpha_{ic})}{\mathcal{B}(\alpha_{ic}, \alpha_{i0} - \alpha_{ic})} \int \frac{1}{\mathcal{B}(\alpha_{ic} + 1, \alpha_{i0} - \alpha_{ic})} \rho_c^{\alpha_{ic}}(1 - \rho_c)^{\alpha_{i0}-\alpha_{ic}-1} d\rho_c \\
&= \frac{\mathcal{B}(\alpha_{ic} + 1, \alpha_{i0} - \alpha_{ic})}{\mathcal{B}(\alpha_{ic}, \alpha_{i0} - \alpha_{ic})} \cdot 1 \\
&= \frac{\Gamma(\alpha_{ic} + 1)\Gamma(\alpha_{i0})}{\Gamma(\alpha_{i0} + 1)\Gamma(\alpha_{ic})} \\
&= \frac{\alpha_{ic}\Gamma(\alpha_{ic})\Gamma(\alpha_{i0})}{\alpha_{i0}\Gamma(\alpha_{i0})\Gamma(\alpha_{ic})} = \frac{\alpha_{ic}}{\sum_{k=1}^{C} \alpha_{ik}} = \frac{g(f_c(\boldsymbol{x}_i, \boldsymbol{\theta}))}{\sum_{k=1}^{C} g(f_k(\boldsymbol{x}_i, \boldsymbol{\theta}))} \\
&= \mathbb{E}[Dir(\rho_c|\boldsymbol{\alpha}_i)].
\end{aligned}
\tag{12}
\tag{13}
$$

Specially, if $g(\cdot)$ adopts the exponential function, traditional softmax-based models can be viewed as predicting the expectation of Dirichlet distribution.

## E.2 Expected Entropy $\mathbb{E}_{p(\boldsymbol{\rho}|\boldsymbol{x}_j,\boldsymbol{\theta})}[H[P(y|\boldsymbol{\rho})]]$

Given sample $\boldsymbol{x}_j$ and model parameters $\theta$, the corresponding expected entropy $\mathbb{E}_{p(\boldsymbol{\rho}|\boldsymbol{x}_j,\boldsymbol{\theta})}[H[P(y|\boldsymbol{\rho})]]$ is formulated as

$$
\begin{aligned}
\mathbb{E}_{p(\boldsymbol{\rho}|\boldsymbol{x}_j,\boldsymbol{\theta})}[H[P(y|\boldsymbol{\rho})]] &= \mathbb{E}_{p(\boldsymbol{\rho}|\boldsymbol{x}_j;\boldsymbol{\theta})}[-\sum_{c=1}^{C}\rho_c\, log\, \rho_c] \\
&= -\sum_{c=1}^{C}\mathbb{E}_{p(\boldsymbol{\rho}|\boldsymbol{x}_j;\boldsymbol{\theta})}[\rho_c\, log\, \rho_c] \\
&= -\sum_{c=1}^{C}\int p(\boldsymbol{\rho}|\boldsymbol{x}_j;\boldsymbol{\theta})\rho_c\, log\, \rho_c\, d\boldsymbol{\rho} \\
&= -\sum_{c=1}^{C}\int\int\cdots\int p(\rho_1,\rho_2,\cdots,\rho_C|\boldsymbol{x}_j;\boldsymbol{\theta})\rho_c\, log\, \rho_c\, d\rho_1 d\rho_2\cdots d\rho_C \\
&= -\sum_{c=1}^{C}\int(\rho_c log\rho_c)(\int\cdots\int\int\cdots\int p(\rho_1,\rho_2,\cdots,\rho_C|\boldsymbol{x}_j;\boldsymbol{\theta})\, d\rho_1\cdots d\rho_{c-1}d\rho_{c+1}\cdots d\rho_C)d\rho_c \\
&= -\sum_{c=1}^{C}\mathbb{E}_{p(\rho_c|\boldsymbol{x}_j;\boldsymbol{\theta})}[\rho_c\, log\, \rho_c]. \quad\quad (14)
\end{aligned}
$$

Combining the probability density function in Eq. (11), we can further derive $\mathbb{E}_{p(\rho_c|\boldsymbol{x}_j;\boldsymbol{\theta})}[\rho_c\, log\, \rho_c]$ as

$$
\begin{aligned}
\mathbb{E}_{p(\rho_c|\boldsymbol{x}_j;\boldsymbol{\theta})}[\rho_c\, log\, \rho_c] &= \int(\rho_c log\rho_c)\frac{1}{\mathcal{B}(\alpha_{jc},\alpha_{j0}-\alpha_{jc})}\rho_c^{\alpha_{jc}-1}(1-\rho_c)^{\alpha_{j0}-\alpha_{jc}-1}\, d\rho_c \\
&= \frac{\mathcal{B}(\alpha_{jc}+1,\alpha_{j0}-\alpha_{jc})}{\mathcal{B}(\alpha_{jc},\alpha_{j0}-\alpha_{jc})}\int(log\rho_c)\frac{1}{\mathcal{B}(\alpha_{jc}+1,\alpha_{j0}-\alpha_{jc})}\rho_c^{\alpha_{jc}}(1-\rho_c)^{\alpha_{j0}-\alpha_{jc}-1}\, d\rho_c \\
&= \frac{\Gamma(\alpha_{jc}+1)\Gamma(\alpha_{j0})}{\Gamma(\alpha_{j0}+1)\Gamma(\alpha_{jc})}\mathbb{E}_{\rho_c\sim Beta(\rho_c|\alpha_{jc}+1,\alpha_{j0}-\alpha_{jc})}[log\rho_c] \quad\quad (15) \\
&= \frac{\alpha_{jc}}{\alpha_{j0}}\left(\psi(\alpha_{jc}+1)-\psi(\alpha_{j0}+1)\right), \quad\quad (16)
\end{aligned}
$$

where $\psi(\cdot)$ is the digamma function, $\alpha_{jc}$ is the $c$-th element of vector $\boldsymbol{\alpha}_j$ and $\alpha_{j0}=\sum_{k=1}^{C}\alpha_{jk}$.

Finally, the expected entropy for sample $\boldsymbol{x}_j$ is denoted as

$$
\begin{aligned}
\mathbb{E}_{p(\boldsymbol{\rho}|\boldsymbol{x}_j,\boldsymbol{\theta})}[H[P(y|\boldsymbol{\rho})]] &= -\sum_{c=1}^{C}\mathbb{E}_{p(\rho_c|\boldsymbol{x}_j;\boldsymbol{\theta})}[\rho_c\, log\, \rho_c] \\
&= -\sum_{c=1}^{C}\frac{\alpha_{jc}}{\alpha_{j0}}\left(\psi(\alpha_{jc}+1)-\psi(\alpha_{j0}+1)\right) \\
&= \sum_{c=1}^{C}\bar{\rho}_{jc}\left(\psi(\sum_{k=1}^{C}\alpha_{jk}+1)-\psi(\alpha_{jc}+1)\right), \quad\quad (17)
\end{aligned}
$$

where $\bar{\rho}_{jc}=\frac{\alpha_{jc}}{\alpha_{j0}}=\mathbb{E}[Dir(\rho_c|\boldsymbol{\alpha}_j)]$.

## E.3 Mutual Information $I[y,\boldsymbol{\rho}|\boldsymbol{x}_j,\boldsymbol{\theta}]$

According to the definition of mutual information (Kieffer, 1994; Shannon, 1948), $I[y,\boldsymbol{\rho}|\boldsymbol{x}_j,\boldsymbol{\theta}]$ can be expressed as

$$
I[y,\boldsymbol{\rho}|\boldsymbol{x}_j,\boldsymbol{\theta}] = \int\sum_{c=1}^{C}p(y=c,\boldsymbol{\rho}|\boldsymbol{x}_j,\boldsymbol{\theta})\, log\, \frac{p(y=c,\boldsymbol{\rho}|\boldsymbol{x}_j,\boldsymbol{\theta})}{p(y=c|\boldsymbol{x}_j,\boldsymbol{\theta})p(\boldsymbol{\rho}|\boldsymbol{x}_j,\boldsymbol{\theta})}d\boldsymbol{\rho}. \quad\quad (18)
$$

Since the deep model induces the Markov chain $(\boldsymbol{x}_j,\boldsymbol{\theta})\to\boldsymbol{\rho}\to y$, we have $y$ and $(\boldsymbol{x}_j,\boldsymbol{\theta})$ conditionally independent given $\boldsymbol{\rho}$, i.e., $p(y,\boldsymbol{\rho}|\boldsymbol{x}_j,\boldsymbol{\theta})=p(y|\boldsymbol{\rho})p(\boldsymbol{\rho}|\boldsymbol{x}_j,\boldsymbol{\theta})$. Then, Eq. (18) can be further

derived as

$$
\begin{aligned}
I[y, \boldsymbol{\rho}|\boldsymbol{x}_j, \boldsymbol{\theta}] &= \int p(\boldsymbol{\rho}|\boldsymbol{x}_j, \boldsymbol{\theta}) \sum_{c=1}^{C} p(y=c|\boldsymbol{\rho}) \, log \, \frac{p(y=c|\boldsymbol{\rho})}{p(y=c|\boldsymbol{x}_j, \boldsymbol{\theta})} d\boldsymbol{\rho} \\
&= \int p(\boldsymbol{\rho}|\boldsymbol{x}_j, \boldsymbol{\theta}) \sum_{c=1}^{C} \big( \rho_c \, log \, \rho_c - \rho_c \, log \, p(y=c|\boldsymbol{x}_j, \boldsymbol{\theta}) \big) d\boldsymbol{\rho} \\
&= \int p(\boldsymbol{\rho}|\boldsymbol{x}_j, \boldsymbol{\theta}) \sum_{c=1}^{C} (\rho_c \, log \, \rho_c) d\boldsymbol{\rho} - \int p(\boldsymbol{\rho}|\boldsymbol{x}_j, \boldsymbol{\theta}) \sum_{c=1}^{C} (\rho_c \, log \, \frac{\alpha_{jc}}{\sum_{k=1}^{C} \alpha_{jk}}) d\boldsymbol{\rho} \quad (19) \\
&= \mathbb{E}_{p(\boldsymbol{\rho}|\boldsymbol{x}_j, \boldsymbol{\theta}) \sim Dir(\boldsymbol{\rho}|\boldsymbol{\alpha}_j)} [\sum_{c=1}^{C} \rho_c \, log \, \rho_c] - \sum_{c=1}^{C} \big( log \frac{\alpha_{jc}}{\sum_{k=1}^{C} \alpha_{jk}} \big) \mathbb{E}_{p(\boldsymbol{\rho}|\boldsymbol{x}_j, \boldsymbol{\theta}) \sim Dir(\boldsymbol{\rho}|\boldsymbol{\alpha}_j)} [\rho_c] \\
&= \mathbb{E}_{p(\boldsymbol{\rho}|\boldsymbol{x}_j, \boldsymbol{\theta}) \sim Dir(\boldsymbol{\rho}|\boldsymbol{\alpha}_j)} [\sum_{c=1}^{C} \rho_c \, log \, \rho_c] - \sum_{c=1}^{C} \bar{\rho}_{jc} log \bar{\rho}_{jc} \\
&= \sum_{c=1}^{C} \bar{\rho}_{jc} \Big( \psi(\alpha_{jc}+1) - \psi(\sum_{k=1}^{C} \alpha_{jk}+1) \Big) - \sum_{c=1}^{C} \bar{\rho}_{jc} log \bar{\rho}_{jc}. \quad (20)
\end{aligned}
$$

The derivation of Eq. (19) is based on the conclusion from Eq. 12, i.e., $P(y=c|\boldsymbol{x}_j, \boldsymbol{\theta}) = \frac{\alpha_{jc}}{\sum_{k=1}^{C} \alpha_{jk}}$. And Eq. 20 is based on the conclusion in Section E.2.

### E.4    KULLBACK-LEIBLER DIVERGENCE $\mathcal{L}_{kl}$

For $p(\boldsymbol{\rho}|\tilde{\boldsymbol{\alpha}}_i) \sim Dir(\boldsymbol{\rho}|\tilde{\boldsymbol{\alpha}}_i)$, its probability density function is defined as

$$
p(\boldsymbol{\rho}|\tilde{\boldsymbol{\alpha}}_i) = \frac{1}{\mathfrak{B}(\tilde{\boldsymbol{\alpha}}_i)} \prod_{c=1}^{C} \rho_c^{\tilde{\alpha}_{ic}-1}, \quad (21)
$$

where $\mathfrak{B}(\cdot)$ is the multivariate Beta function, $\mathfrak{B}(\tilde{\boldsymbol{\alpha}}_i) = \frac{\prod_{c=1}^{C} \Gamma(\tilde{\alpha}_{ic})}{\Gamma(\sum_{c=1}^{C} \tilde{\alpha}_{ic})}$ and $\Gamma(\cdot)$ is the Gamma function. The Kullback-Leibler Divergence between Dirichlet distribution $Dir(\boldsymbol{\rho}|\tilde{\boldsymbol{\alpha}}_i)$ and $Dir(\boldsymbol{\rho}|\mathbf{1})$ is formulated as

$$
\begin{aligned}
KL\big[Dir(\boldsymbol{\rho}|\tilde{\boldsymbol{\alpha}}_i) \| Dir(\boldsymbol{\rho}|\mathbf{1})\big] &= \int p(\boldsymbol{\rho}|\tilde{\boldsymbol{\alpha}}_i) \, log \, \frac{p(\boldsymbol{\rho}|\tilde{\boldsymbol{\alpha}}_i)}{p(\boldsymbol{\rho}|\mathbf{1})} d\boldsymbol{\rho} \\
&= \int \Big( \frac{1}{\mathfrak{B}(\tilde{\boldsymbol{\alpha}}_i)} \prod_{c=1}^{C} \rho_c^{\tilde{\alpha}_{ic}-1} \Big) log \Big( \frac{\mathfrak{B}(\mathbf{1})}{\mathfrak{B}(\tilde{\boldsymbol{\alpha}}_i)} \prod_{c=1}^{C} \rho_c^{\tilde{\alpha}_{ic}-1} \Big) d\boldsymbol{\rho} \\
&= log \frac{\mathfrak{B}(\mathbf{1})}{\mathfrak{B}(\tilde{\boldsymbol{\alpha}}_i)} \int (\frac{1}{\mathfrak{B}(\tilde{\boldsymbol{\alpha}}_i)} \prod_{c=1}^{C} \rho_c^{\tilde{\alpha}_{ic}-1}) \, d\boldsymbol{\rho} + \int (log \prod_{c=1}^{C} \rho_c^{\tilde{\alpha}_{ic}-1})(\frac{1}{\mathfrak{B}(\tilde{\boldsymbol{\alpha}}_i)} \prod_{c=1}^{C} \rho_c^{\tilde{\alpha}_{ic}-1}) \, d\boldsymbol{\rho} \\
&= log \frac{\mathfrak{B}(\mathbf{1})}{\mathfrak{B}(\tilde{\boldsymbol{\alpha}}_i)} \cdot 1 + \mathbb{E}_{\boldsymbol{\rho} \sim Dir(\boldsymbol{\rho}|\tilde{\boldsymbol{\alpha}}_i)} [log \prod_{c=1}^{C} \rho_c^{\tilde{\alpha}_{ic}-1}] \\
&= log \frac{\mathfrak{B}(\mathbf{1})}{\mathfrak{B}(\tilde{\boldsymbol{\alpha}}_i)} + \sum_{c=1}^{C} (\tilde{\alpha}_{ic}-1) \mathbb{E}_{\rho_c \sim Beta(\rho_c|\tilde{\alpha}_{ic}, \tilde{\alpha}_{i0}-\tilde{\alpha}_{ic})} [log \rho_c] \\
&= log \Big( \frac{\Gamma(\sum_{c=1}^{C} \tilde{\alpha}_{ic})}{\Gamma(C) \prod_{c=1}^{C} \Gamma(\tilde{\alpha}_{ic})} \Big) + \sum_{c=1}^{C} (\tilde{\alpha}_{ic}-1) \Big[ \psi(\tilde{\alpha}_{ic}) - \psi(\sum_{k=1}^{C} \tilde{\alpha}_{ik}) \Big]. \quad (22)
\end{aligned}
$$

Thus, the computable expression of $\mathcal{L}_{kl}$ is given by

$$\mathcal{L}_{kl} = \frac{1}{C \cdot n_s} \sum_{\boldsymbol{x}_i \in \mathcal{S}} KL\big[Dir(\boldsymbol{\rho}|\tilde{\boldsymbol{\alpha}}_i)\|Dir(\boldsymbol{\rho}|\mathbf{1})\big] + \frac{1}{C \cdot |\mathcal{T}^l|} \sum_{\boldsymbol{x}_j \in \mathcal{T}^l} KL\big[Dir(\boldsymbol{\rho}|\tilde{\boldsymbol{\alpha}}_j)\|Dir(\boldsymbol{\rho}|\mathbf{1})\big],$$

$$= \frac{1}{C \cdot n_s} \sum_{\boldsymbol{x}_i \in \mathcal{S}} log\left(\frac{\Gamma(\sum_{c=1}^{C} \tilde{\alpha}_{ic})}{\Gamma(C)\prod_{c=1}^{C} \Gamma(\tilde{\alpha}_{ic})}\right) + \sum_{c=1}^{C} (\tilde{\alpha}_{ic} - 1)\left[\psi(\tilde{\alpha}_{ic}) - \psi(\sum_{k=1}^{C} \tilde{\alpha}_{ik})\right]$$

$$+ \frac{1}{C \cdot |\mathcal{T}^l|} \sum_{\boldsymbol{x}_j \in \mathcal{T}^l} log\left(\frac{\Gamma(\sum_{c=1}^{C} \tilde{\alpha}_{jc})}{\Gamma(C)\prod_{c=1}^{C} \Gamma(\tilde{\alpha}_{jc})}\right) + \sum_{c=1}^{C} (\tilde{\alpha}_{jc} - 1)\left[\psi(\tilde{\alpha}_{jc}) - \psi(\sum_{k=1}^{C} \tilde{\alpha}_{jk})\right]. \quad (23)$$

