# OpenReview forum: "Dirichlet-based Uncertainty Calibration for Active Domain Adaptation"
_ICLR.cc/2023/Conference — ICLR 2023 notable top 25%_

### Official Review · Reviewer_Rp4n · 2022-10-20

**Confidence:** 4
**Correctness:** 4
**Technical Novelty And Significance:** 4
**Empirical Novelty And Significance:** 3
**Recommendation:** 8

**Clarity, Quality, Novelty And Reproducibility:**

The paper is well written.  The distinction between data and distributional uncertainty to enhance EDL training for active learning in domain adaptation and for selecting the labeled sets is novel and significant.  The experimentation in terms of baselines and two application areas (image classification and semantic segmentation) is very strong.  The explanation of the approach is detailed enough to reproduce the work.

Perhaps once could quibble that the KL regularization term in EL does not appear to be annealed in (8).  In training EDL, the annealing is important as EDL must learn to discriminate before it can calibrate evidence. Perhaps for the intermediate steps in active learning to select samples to label, this is not critical.

Regarding other Dirichlet-based single pass methods, the  authors argue that Dirichlet Prior Network requires specification of an OOD dataset, whereas, EDL does not.  However, EDL training may not properly capture the epistemic uncertainty.  Not sure this matter, but this reviewer is curious if Dirchlet Posterior Networks (Posterior Network: Uncertainty Estimation without
OOD Samples via Density-Based Pseudo-Counts, NeurIPS 2020) and/or Generative Evidential Neural Networks (Uncertainty-Aware Deep Classifiers Using Generative Models, AAAI 2020) can provide better results.  Both of these methods do not require one to explicitly provide OOD samples. All of this would be interesting as future work.

**Details Of Ethics Concerns:**

I do not have any ethical concerns.

**Strength And Weaknesses:**

The strength of the paper is that is proposes a novel domain adaptation active learning method that is a significant modification of evidential deep learning in its use of distribution and data uncertainty in its training and its selection of the samples to label.  The experimental validation is done for domain adaptation in image classification and semantic segmentation applications. The experimental results also provides ablation and other studies demonstrating the effectiveness of various components of the proposed DUC method. The weaknesses are fairly minor.  It would be nice if the experimental sections explains that datasets represent different domains to serve as the source and target domains. Perhaps one could quibble that other single-pass Dirichlet-based methods should be studied. Nevertheless, the study is already very extensive.

**Summary Of The Paper:**

This paper proposes a novel domain adaptation active learning scheme building upon evidential deep learning. Namely, it computes distributional (or epistemic) uncertainty as mutual information and data (or epistemic) uncertainty as the expected entropy.  Then it uses a two stage approach that selects  to label the largest data uncertainty samples from a pool of the largest distribution uncertainty samples. The EDL method is enhanced to minimize data and distribution uncertainty for the newly labeled target samples.  This helps to identify remaining gaps in the labeling of the target samples.  The paper demonstrates via extensive experiments that this new Dirichlet-based Uncertainty Calibration method is able to find a better set of target samples to label than state-of-the-art active learning and domain adaptation active learning methods.

**Summary Of The Review:**

The paper proposes a novel approach that is sensible and validated in extensive experiments spanning two different applications.

I have looked over the revised paper and the authors' response.  I think this is a good paper that should be accepted, and my recommendation reflects this.  I think the other reviewers may not have appreciated the significance of the changes to EDL so that it can be applied for active domain adaptation. I can attest that the contribution of the paper in that regard is significant. I also believe that the authors have significantly enhanced the experiments in response to the other reviewers. The experimental section is comprehensive and demonstrates an advancement relative to the SOTA.

---

### Official Review · Reviewer_arzy · 2022-10-25

**Confidence:** 4
**Correctness:** 4
**Technical Novelty And Significance:** 3
**Empirical Novelty And Significance:** 3
**Recommendation:** 6

**Clarity, Quality, Novelty And Reproducibility:**

The idea is not novel by itself , as the authors also mention it has been used for OOD detection. However,  it may to be novel in the context of active learning domain adaptation which makes sense.

**Strength And Weaknesses:**


Strengths
- simple yet effective method
- active domain adaptation is a practical approach for domain adaptation. Probably you can improve results by incorporating ideas from semi-supervised learning to use the unlabeled data in training as well.
- extensive experiments and calibration analysis
- ablation studies are good

Weaknesses:
- there are no error bars on the results of the experiments. Its important to have error bars to know if the improved results are statistically significant .
- I think experimenting on more datasets would add merit to the paper and give more credibility to the approach. large scale datasets like Vista could be good additional datasets.

**Summary Of The Paper:**

The authors propose an uncertainty aware active learning domain adaptation based on imposing a dirichlet prior over model predictions. This allows them to get a distribution over model predictions, which in turn enables them to compute 2 kinds of uncertainty measures for each target datapoint. 1) target distribution uncertainty 2) data complexity uncertainty.
Using these two measures of uncertainty the active learning is done in 2 steps, first they select data points with highest amount of target distribution uncertainty and then from those datapoints with highest data uncertainty are chosen for being assigned ground truth labels.
The results on 2 classification datasets and 2 segmentation datasets show promising results.


**Summary Of The Review:**

I think the novelty is limited in the theoretical sense but its been used in an interesting application. That being said, the experiments are sound and do confirm the merits of the approach.

---

### Official Review · Reviewer_CH6i · 2022-10-25

**Confidence:** 4
**Correctness:** 4
**Technical Novelty And Significance:** 2
**Empirical Novelty And Significance:** 2
**Recommendation:** 5

**Clarity, Quality, Novelty And Reproducibility:**

The paper is clear and nicely structured to read. The novelty side of the work is limited.
Fig 1a: the color bar is not clear, even the description of Fig 1b is very short for readers to clearly understand the problem and what needs to be done.


**Strength And Weaknesses:**

Strengths:

Active domain adaptation is a new sub-track of domain adaptation and the authors have proposed a new DUC approach to solving it.
The authors dealing with this use of Dirichlet prior and probability space on simplex mitigate the issues of the point estimate and consider all possible predictions.
The authors provide a detailed analysis of their approach with the intuition behind the optimization.
The results on miniDomainNet and Office-Home show improvement over the state-of-the-art (SOTA) models. This is true for GTAV and SYNTHIA experiments as well.
The ablation study in Table 5 shows that having loss of distribution uncertainty and loss of data uncertainty with the selection process helped in improving other SOTA counterparts.
Also, the authors studied the effect of different first-round selection ratios κ% on the Office-Home dataset which remains almost the same except for k=1% , where it was lowest (Fig 4a).

Weaknesses:
Two main contributions in this work are derived from previous works:
	1: Predictive uncertainty estimation via prior networks (Malinin & Gales, 2018).
	2: Evidential deep learning to quantify classification uncertainty (Sensoy et al., 2018).
In my opinion, the above two prior work have been put together in this area of active DA.

Why has the author not reported the result on the VisDA-2017 dataset used in the EADA paper, since EADA is the current SOTA model? Why has the author not included the EADA paper as a baseline for the GTAV dataset in Table 3?
It is not reported in the paper where these numbers in Table 1 for EADA on miniDomainet have come from.
It is not reported in the paper what is the performance gap between Active DA methods and the “Full Supervised" on the target domain.



**Summary Of The Paper:**

The authors propose a solution to the problem of active domain adaptation, where limited target data is annotated to maximally benefit the model adaptation. The proposed solution called "Dirichlet-based Uncertainty Calibration (DUC)" achieves both target representativeness and mitigation of miscalibration. The authors place a Dirichlet prior on the class probabilities which helps interpret it as the distribution over the probability simplex. There is a two-round selection strategy, uncertainty of distribution and uncertainty of data, to select the target samples for annotation. The authors show the superiority of their approach on cross-domain image classification and semantic segmentation tasks.

**Summary Of The Review:**

Overall, the paper is well written, and the results show improvement over the baseline models. However, I feel the technical contribution and the novelty are limited.

---

### Decision · Program_Chairs · 2023-01-20

**Decision:**

Accept: notable-top-25%

**Justification For Why Not Higher Score:**

The proposed approach combines some previous techniques to address a new problem, but the base techniques that are combined are not novel.

**Justification For Why Not Lower Score:**

Active domain adaptation is a relatively new subfield of domain adapatation.  Very little work has been done in this subfield.  The proposed approach makes an important advance that will be of high interest to the community.

**Metareview: Summary, Strengths And Weaknesses:**

The paper considers a relatively new problem: active domain adaptation.  The proposed approach leverages previous work about uncertainty quantification in traditional active learning and extends it to domain adaptation.  The resulting approach is quite effective and represents an important advance for active domain adaptation.

Strengths:
* Paper tackles a relatively new problem: active domain adapatation
* Resulting approach is quite effective and represents an important advance for active domain adaptation

Weaknesses:
* The proposed method combines existing methods from traditional active learning.  However, the extension leads to a novel solution for active domain adaptation that represents an important advance

**Note From Pc:**

if the above contains the word "oral" or "spotlight" please see: "oral" presentation means -> notable-top-5% and "spotlight" means -> notable-top-25%. As stated in our emails, we are disassociating presentation type from AC recommendations